# SOA yields from $C_{10}$ alkanes and oxygenates

Frans Graeffe[1], Kalle Kupi[1], Hilkka Timonen[2] and Mikael Ehn[1]

[1]Institute for Atmospheric and Earth System Research/Physics, Faculty of Science, University of Helsinki, Helsinki, 00014, Finland

[2]Atmospheric Composition Research, Finnish Meteorological Institute, Helsinki, Finland

*Correspondence to*: Frans Graeffe (frans.graeffe@helsinki.fi) and Mikael Ehn (mikael.ehn@helsinki.fi)

**Abstract.**

Alkanes are hydrocarbons that are emitted into the atmosphere mainly by human activities such as combustion processes and via the use of volatile chemical products (VCPs). They are an important group of volatile organic compounds (VOCs) that can produce secondary organic aerosol (SOA) in the atmosphere via hydroxyl (OH) radical initiated reactions. For many other compound groups (e.g., aromatics and monoterpenes), highly oxygenated organic molecules (HOMs) formed via autoxidation have been shown to be an important link between VOCs and SOA. Although alkane SOA has been intensively studied over the last decades, the importance of autoxidation and HOM in this system has received limited attention. The first HOM observations were only recently reported, but their relation so SOA has not been directly studied. Here, we show results of SOA yields from seven $C_{10}$ alkanes and their oxygenated derivatives in oxidation flow reactor experiments. We observe the well-known behaviour of increased SOA yield with different structure in the order of cyclic > linear > branched. We also measured HOMs, and all seven SOA precursors produced detectable amounts of products, but HOM quantification was not possible due to the experimental setup configuration focusing on SOA formation. However, a comparison to previously reported HOM yields for the same precursors was conducted, indicating a correlation between HOM and SOA yields. Although not quantifiable, our own HOM observations did indicate that multi-generation OH oxidation played an important role in the SOA formation in our study.

## 1 Introduction

Alkanes are a major part of anthropogenic VOC (AVOC) emissions, especially in urban areas (Leuchner and Rappenglück, 2010; Garzón et al., 2015; Song et al., 2019; Gu et al., 2021). They mainly originate from combustion processes and vehicle exhaust (Jathar et al., 2017; Huang et al., 2018) as well as from volatile chemical products (VCPs), including adhesives, sanitiser, coatings and personal care products (Mcdonald et al., 2018; Wang et al., 2024). With the decrease of the traditional anthropogenic VOC emissions, including the fossil fuel emissions from transportation/tailpipe emissions (Bishop and Haugen,

2018), other emissions can increase in relative importance, for example VCPs (Mcdonald et al., 2018; Coggon et al., 2021). As alkanes and their oxygenates are also part of VCPs, we need to understand better how they affect the air quality and SOA. The importance of VCP emissions have gotten more attention, and their significant contribution to urban emissions has only recently been addressed in more detail. Gu et al. (2021) found that alkanes (from e.g. mineral spirits) and aromatics were the

major contributor to SOA-forming potential in Los Angeles county, USA, while Seltzer et al. (2021) found VCPs (including linear, cyclic and branched alkanes) a significant source of AVOC emissions in the USA. This demonstrates that more studies for these compounds are needed, as older measurements might not represent current AVOC emissions and compound specific distributions anymore.

Numerous previous studies have shown that alkanes are capable of producing significant amounts of SOA and the SOA yield

depends among other things on the alkane carbon number and structure (Lambe et al., 2012; Tkacik et al., 2012; Hunter et al., 2014; Li et al., 2019; Hallward-Driemeier et al., 2024; Madhu et al., 2024; Jo et al., 2024).

Recently, Wang et al. (2021) showed that alkanes can undergo autoxidation more efficiently than previously thought, and have the potential to produce highly oxidized products, including those with more than six O-atoms, which are often labelled highly oxygenated organic molecules (HOM, (Bianchi et al., 2019)). Previous studies has shown that HOM production (via

autoxidation) is an important link between VOCs and SOA for many systems, such as in the oxidation of monoterpene or aromatics (Ehn et al., 2014; Garmash et al., 2020). Wang et al. (2021) did not only measure HOM yields, but showed that the oxygen content in oxidation products generally increased when more peroxy radicals ($RO_2$) were converted to alkoxy radicals (RO), even though not always reaching six or more O-atoms. Much of the O-atom incorporation was attributed to $RO_2$ reactions with other $RO_2$ radicals or NO, forming RO able to isomerize and thus allow reactions with molecular $O_2$. This is in contrast

to many monoterpenes where the $RO_2$ radicals themselves can undergo isomerization reactions (autoxidation), owing to suitable structures in the monoterpene-derived radicals which are less common in alkanes.

In this study, we aim to extend the work of Wang et al. (2021) by measuring SOA yields for many of the alkanes and oxygenates that they reported, in order to assess the links between HOM and SOA formation. We produced SOA (mass concentrations ranging from 3 to 66 µg m$^{-3}$) from 7 different $C_{10}$-VOCs (alkanes and their oxygenated derivatives) to investigate their SOA

yields. The experiments were done in an oxidation flow reactor, in the absence of $NO_x$ and seed particles, simulating fresh SOA. Although alkanes, and other AVOCs, are mainly emitted in the presence of $NO_x$, they can be transported downwind to low-$NO_x$ environments, motivating our study similarly as in Li et al. (2019). In addition, VCPs (including sanitizers and adhesives) are often used indoors (low-$NO_x$ conditions). Furthermore, Hallward-Driemeier et al. (2024) showed that five $C_{10}$ alkanes and oxygenated derivatives exhibits in general higher SOA yields at lower $NO_x$ concentrations, indicating that low-

$NO_x$ conditions can be seen as an upper-limit of alkane-SOA yields. Lastly, the comparison of only low-$NO_x$ versus high-$NO_x$ conditions is not strictly simple (Wennberg, 2024), and measuring SOA yields at different $NO_x$ concentrations would be out of scope for this study. Therefore, we are confident that our SOA yields in the absence of $NO_x$ is both useful and representative for the conditions described above. We also measured HOMs at the exit of the flow reactor, but the experimental setup

(optimized for SOA formation) did not allow direct quantification of the HOM yields or new detailed insights in alkane HOM

formation, except for assessing the role of multi-generational OH oxidation.

## 2 Methods

The experimental setup used in this work is presented in Fig. A1, and Table 1 summarises all the used VOCs. The

instrumentation is described in the sections below.

**Table 1. Summary of all the VOCs (including name, molecular formula, structure and rate constant with OH), amount of injected and reacted VOC, formed SOA mass and their SOA yields.**

| Compound | Molecular formula | Structure | Rate constant k ($cm^3$ molecule$^{-1}$ s$^{-1}$) | Injected VOC (ppb) | Reacted VOC (ppb) | Formed SOA-mass ($\mu g\ m^{-3}$) | SOA yield (%) |
|---|---|---|---|---|---|---|---|
| **Alkane** | | | | | | | |
| n-decane | $C_{10}H_{22}$ | | $1.1\times10^{-11}$ [a] | 39-98 | 19-29 | 4.8-9.3 | 4.4-5.5 |
| 2,7-dimethyloctane | $C_{10}H_{22}$ | | $1.1\times10^{-11}$ [b] | 29-120 | 16-32 | 2.5-8.4 | 2.8-5.2 |
| n-butylcyclohexane | $C_{10}H_{20}$ | | $1.47\times10^{-11}$ [c] | 14-110 | 10-33 | 11-39 | 17-23 |
| cis-decalin | $C_{10}H_{18}$ | | $2.01\times10^{-11}$ [d] | 16-79 | 12-30 | 22-66 | 32-39 |
| **Oxygenate** | | | | | | | |
| decanal | $C_{10}H_{20}O$ | | $3.25\times10^{-11}$ [e] | 13-52 | 11-27 | 4.1-26 | 6.0-15 |
| 2-decanone | $C_{10}H_{20}O$ | | $1.32\times10^{-11}$ [f] | 43-120 | 21-33 | 4.4-20 | 3.4-10 |
| 1-decanol | $C_{10}H_{22}O$ | | $1.5\times10^{-11}$ [g] | 120-360 | 33-41 | 7.8-24 | 3.6-9.1 |

(a) Atkinson (2003)

(b) estimate (Parchem, 2025)

(c) Atkinson and Arey (2003)

(d) Atkinson et al. (1983)

(e) estimate, average from Wang et al. (2021) and Pubchem (2025b)

(f) Wallington and Kurylo (1987)

(g) estimate (Wang et al., 2021; Pubchem, 2025a)

## 2.1 SOA production with a PAM

We used a Potential Aerosol Mass oxidation flow reactor (PAM-OFR, hereafter PAM) (Kang et al., 2007; Lambe et al., 2011)
to generate SOA via homogeneous nucleation from OH oxidation of the precursors. The PAM chamber is cylindrical and made
of aluminium with a volume of ~13 L. The PAM was operated in OFR185 mode, meaning that UV lamps inside the PAM
emitted at two wavelengths, 185 nm and 254 nm. Ozone ($O_3$) is primarily produced from the photolysis of molecular oxygen:
$O_2 + hv$ (185 nm) $\rightarrow$ 2 $O(^3P)$, followed by $O_2 + O(^3P) \rightarrow O_3$. Hydroxyl radicals (OH) are primarily produced from $O_3 + hv$
(254 nm) $\rightarrow O_2 + O(^1D)$ followed by $O(^1D) + H_2O \rightarrow 2$ OH.
The total flow through the PAM was 10 L $min^{-1}$, consisting of purified air (clean air generator AADCO, series 737-14, Ohio,
USA) that was directed through a water bath to increase the relative humidity (RH = 22 % ± 2 %) and a precursor gas flow
($N_2$ as carrier gas). The residence time was therefore ~80 s. The VOC was continuously injected to the $N_2$ carrier gas flow with
a syringe pump with different injection speeds to get different concentrations of the precursor to the PAM.

The flows, RH, and voltages of the UV lamps (100 V) were kept constant through all experiments, only the VOC concentration
was changing.

The integrated OH exposure was estimated by the estimation equation presented in Li et al. (2016), taking into account different
SOA precursors and concentrations (with different reactivity toward OH and different external OH reactivity, $OH_{ext}$).

The integrated OH exposure ($OH_{exp}$) varied between $0.8 \times 10^{10}$ and $8.6 \times 10^{10}$ molecules $cm^{-3}$ s in all of the experiments.
Assuming atmospheric OH concentration of $1.5 \times 10^6$ molecules $cm^{-3}$, our experiments are equivalent to approximately 1 to 16
hours of atmospheric aging, i.e., fresh SOA.

Prior the actual measurements, we used a PTR-ToF (described in the section below) to measure $OH_{exp}$ by measuring the decay
of a compound at different UV intensities, as described in Lambe et al. (2011). Instead of using $SO_2$, the traditional choice, we
used nonanal as it represents better the group of compounds used in this study. The measured $OH_{exp}$ (at 35 ppb of nonanal, OH
+ nonanal rate constant was $3.6 \times 10^{-11}$ $cm^3$ $molecule^{-1}$ $s^{-1}$ from Bowman et al. (2003)) was $5.5 \times 10^{10}$ molecules $cm^{-3}$ s and the
modelled $OH_{exp}$ for the same calibration was $3.5 \times 10^{10}$ molecules $cm^{-3}$ s. However, as we used different precursors and
measured over a wide range of precursor concentrations, and the discrepancy between the two methods were not huge, we
chose to use the model to estimate $OH_{exp}$.

There are several uncertainties with the modelled $OH_{exp}$ which we have taken into account for when using the model, for which
one need to know e.g., residence time, VOC concentration and reaction rate constants. All uncertainties and how they are
applied in the error calculations are described in more detail in Appendix B.

To account for both particle and gas phase losses in the PAM, we have applied two corrections. For measuring particle
transmission at different sizes, we used dry ammonium sulphate particles ranging from 30 nm to 400 nm in diameter (size

selected with a differential mobility analyzer (DMA)). Size dependent particle transmission (Fig. B1) was gained when comparing the particle number concentration before and after the PAM. For particles larger than 80 nm, the transmission was always over 90 %, while it was lower for smaller particle sizes. Correcting for gas phase losses in a system is challenging, however, we applied the model for estimating the fate of low-volatility organic compounds (LVOCs) by Palm et al. (2016). This model account for e.g., wall losses in the PAM and calculates the fraction of LVOCs that are condensed onto particles and forms the SOA we measured with the AMS + SMPS system (described in the section below). In general, over 90% of all LVOC condenses onto particles, with a decreasing trend with increasing $OH_{exp}$ (Fig. B2 as an example output from the model). However, as no seed aerosol is used for these measurements and the model uses condensation sink (CS) as an input parameter, the models does not capture any (wall) losses before nucleation has happened inside the PAM. Therefore, we expect that the real gas phase losses are somewhat greater than the model predicts.

Previous studies (Cappa and Wilson, 2012; Chen et al., 2013) have shown that heterogeneous reactions should not be significant at less than two days of ageing. Therefore, we do not account for any heterogeneous chemistry in our SOA yield calculations as all our measurements were done under one day of atmospheric ageing.

## 2.2 Instrumentation

Both particle and gas phase products were monitored after the PAM. Chemical composition of the freshly produced SOA was measured with a Long Time of Flight Aerosol Mass Spectrometer (LToF-AMS, hereafter AMS) (Aerodyne Research, Inc.). (Decarlo et al., 2006; Graeffe et al., 2023) The AMS was alternating between open and closed mode with 1 min acquisition. Sample flow from the PAM to the AMS was set to 1 L min$^{-1}$ with an external vacuum pump, with only 0.1 L min$^{-1}$ being sampled into the AMS.

In order to bypass the problem of unknown/varying relative ionization efficiency (RIE) and collection efficiency (CE) for our pure organic aerosol in the AMS analysis, we used a custom-built scanning mobility particle sizer (SMPS) in conjunction with the AMS for calculating the SOA mass concentration.

The AMS was used to calculate the elemental ratios (O/C and H/C) of the SOA, while the SMPS measured the particle number size distribution from 10 nm to 500 nm particles. From that, we calculated the volume concentration.

The mass concentration of the SOA was calculated by combining the total particle volume from the SMPS and the SOA density calculated from the elemental ratios. The density was calculated for each step according to the equation in Kuwata et al. (2012), yielding in densities from 1100 to 1400 kg m$^{-3}$. Mass concentration was also corrected by the "LVOC fate correction", according to Palm et al. (2016), as described in the section above.

For measuring HOMs, we used a nitrate chemical ionisation mass spectrometer (NO$_3$-CIMS) (Tofwerk AG/Aerodyne Research, Inc.) (Jokinen et al., 2012) equipped with a LToF and Eisele-type nitrate inlet (Eisele and Tanner, 1993). Our experimental setup was not optimized for the quantifying HOM yields, as our main goal was to study the SOA. The main limitation for the HOM yield quantification was in determining the HOM loss rates. As the NO$_3$-CIMS needs a sample flow

of ~10 L min$^{-1}$ (i.e., sum of all the flows through the PAM), we needed to add a dilution flow directly after the PAM, making sure that all instruments got enough sample flow. With this setup, we lost the majority of the HOMs coming from the PAM before they reached the NO$_3$-CIMS, owing to turbulence and related wall losses in the tubing. In addition, the vast majority of the HOMs produced in the PAM had already condensed onto particles or walls before exiting the PAM. Thus, a higher HOM yield in the PAM may contribute to efficient SOA formation, which, in turn, forms a large sink for the HOM, ultimately decreasing the HOM concentrations. Due to these difficulties, we were not able to estimate the HOM losses and therefore not able to calculate any HOM yields. However, we were able to acquire mass spectra from all of the experiments and the HOM data collected did provide insights on the gas phase reactions.

For VOC measurements, we used a proton transfer reaction ToF mass spectrometer (PTR-ToF 8000, Ionicon Analytik GmbH, Austria)(Jordan et al., 2009). However, the PTR-ToF was operational only during a few individual experiments, including the $OH_{exp}$ calibration, so no detailed results from the actual experiments are shown here.

Ozone concentration was monitored with a photometric O$_3$ analyzer – model 400 (Teledyne API) and RH was kept constant during all experiments, RH = 22 % +- 2 %.

We injected typically 10-120 ppb of precursor VOC to the system, except for 1-decanol as it did not produce enough SOA at those precursor levels. For that, we injected up to 350 ppb to generate higher SOA loadings. The precursor was injected into a N$_2$ carrier gas with a syringe pump.

A typical experiment, showed in Fig. 1, consisted of the following steps: (1) background measurements with no VOC injected in the system, (2) different VOC concentrations injected to the system, producing $0 - 70$ µg m$^{-3}$ of SOA and (3) background measurements.

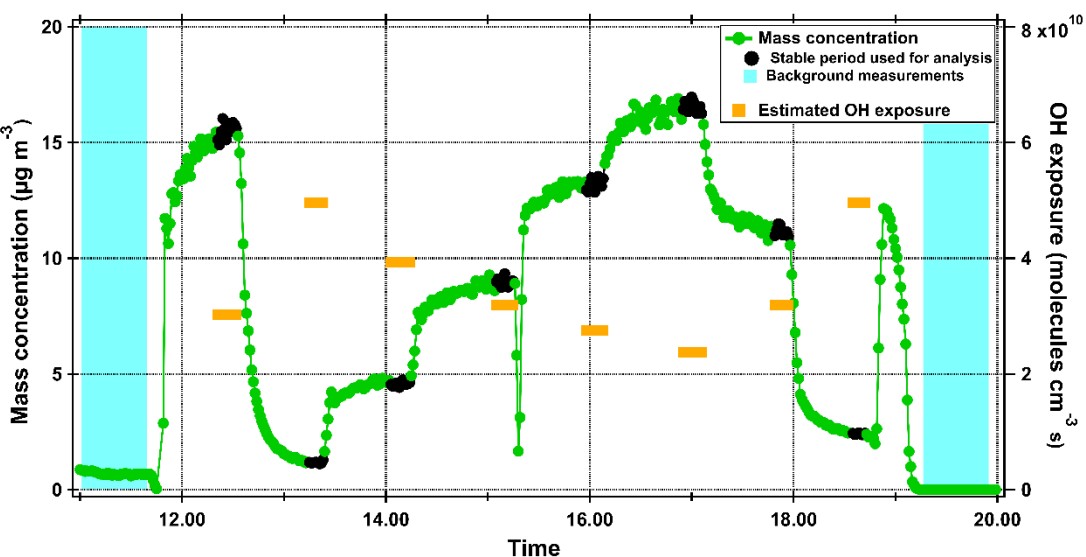

**Figure 1. Time series of 2-decanone SOA mass concentration as an example of a typical experiment. The black points represent the stable time periods that were used for further analysis of a specific precursor/SOA mass concentration. The orange lines are the estimated OH exposures for each stable point and the blue region is the background period prior and after the precursor injections.**

## 3 Results

### 3.1 Particle phase

### 3.1.1 SOA yields

The SOA yields ($Y$) are calculated as the ratio of formed organic aerosol concentration ($C_{OA}$) to reacted precursor concentration ($\Delta VOC$) by:

$$Y = C_{OA}/\Delta VOC \tag{1}$$

The reacted precursor was calculated by:

$$\Delta VOC = [VOC] \times (1 - e^{-k \times OH_{exp}}) \tag{2}$$

, where $k$ is the second-order rate constant of the precursor with OH and [VOC] is the injected VOC concentration (known from the syringe pump).

The amount of reacted precursor varied from 10 to 41 ppb (Fig. 2b), corresponding to 26 % to 86 % of the injected amount of the precursor, expect for 1-decanol, which reacted only 11 % to 28 %.

In Fig. 2 we show the SOA yield as a function of injected VOC, reacted VOC and formed SOA. Summaries of the data from Figure 2 are also presented in Table 1. We can clearly see that the cycloalkanes and decanal had the highest SOA yields, ranging from ~6 % to ~39 %. The lowest SOA yields were from the acyclic alkanes and the other oxygenates (except decanal), ranging from ~3 % to ~10 %.

The SOA mass concentrations ranged from 2.5 to 66 µg m$^{-3}$, which is typical mass concentrations found in the atmosphere. The compounds with high SOA yields, in particular the cycloalkanes, produced SOA efficiently already at low VOC concentrations, as seen in Fig. 2a and b. For example, cis-decalin produced on average over 22 µg m$^{-3}$ of SOA at 10 ppb of reacted precursor, while 1-decanol did not produce any measurable mass below 30 ppb of reacted VOC.

The underlying reason for higher SOA yield for cyclic alkanes has already been discussed in previous studies, e.g., in Lim and Ziemann (2009) and Hunter et al. (2014); during the oxidation step, acyclic compounds have higher risk for fragmentation. Fragmentation of the parent carbon-chain leads to products that are too small and volatile for participating in condensation and SOA formation. During the OH oxidation step, carbon-carbon bond (C-C) scission of linear and branched compounds tends to cause fragmentation, while cyclic compounds can prevent the fragmentation of the parent carbon-chain with ring-openings.

Therefore, cyclic moieties can undergo several functionalization steps including oxygen addition before any major fragmentation occurs.

These results indicate that the oxidation of cycloalkanes produces more efficiently low volatility vapours that can condense and form SOA, compared to acyclic compounds. Furthermore, without seed aerosol, the oxidation products need to have low enough volatility to initiate nucleation already at low precursor concentration. In contrast, the steep increases in SOA yields

for many other precursors in Fig. 2b suggests that partitioning into the aerosol phase is strongly enhanced as the amount of products (and the SOA mass concentrations) increase.

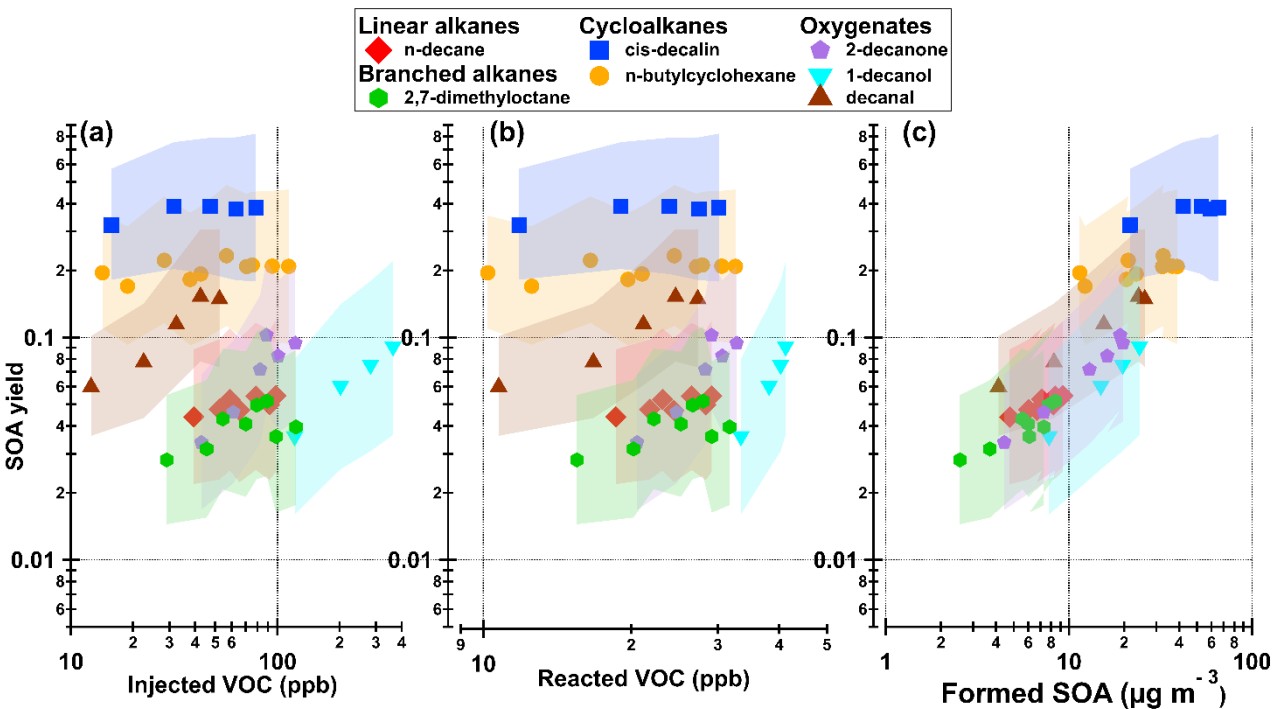

**Figure 2. SOA yield as function of (a) injected VOC, (b) reacted VOC and (c) formed SOA. The shaded area represents the error as described in Appendix B.**

For alkanes, previous studies (Lim and Ziemann, 2005, 2009; Tkacik et al., 2012) have found that the SOA yield increases following the trend cyclic > linear > branched. Although those studies were done in the presence of $NO_x$, while our study was done without any $NO_x$, we find the same overall behaviour for the SOA yields.

For n-decane, Presto et al. (2010) got a SOA yield of 1.5 % at 6 µg m$^{-3}$ of SOA mass (under high $NO_x$ conditions), while we measured a yield of ~5 % at 6 µg m$^{-3}$. In the absence of $NO_x$, n-decane SOA yields have been studied also by Lambe et al. (2012) and Li et al. (2019) in an OFR. However, both studies measured the SOA yields over different $OH_{exp}$, with emphasis on more aged aerosol and the SOA yield evolution with increased $OH_{exp}$. Nevertheless, a SOA yield for n-decane from Lambe et al. (2012) and Li et al. (2019) would be less than 3 % at an equivalent photochemical age of ~0.5 days. Our SOA yields are therefore slightly higher with less ageing.

For cis-decalin, our SOA yield was 32-39 %, while Li et al. (2019) measured a SOA yield of 10 % at similar $OH_{exp}$.

For n-butylcyclohexane, the only reported SOA yield, 38 %, is from Lim and Ziemann (2009) where the experiment was done in presence of both organic seed aerosol and $NO_x$ at over 1800 µg m$^{-3}$ SOA mass concentrations. Their result is therefore not directly comparable to our SOA yield of 17-23 %. However, the presence of seed aerosol can increase the yield up to a factor of 2 to 3 (Lambe et al., 2015; Ahlberg et al., 2019); therefore, our results lie within that range.

For the other precursors used in this study, we are not aware of any previous SOA yield measurements under similar conditions, i.e., in the absence of NO$_x$ and seed aerosol.

### 3.1.2 Van Krevelen diagram and carbon oxidation state

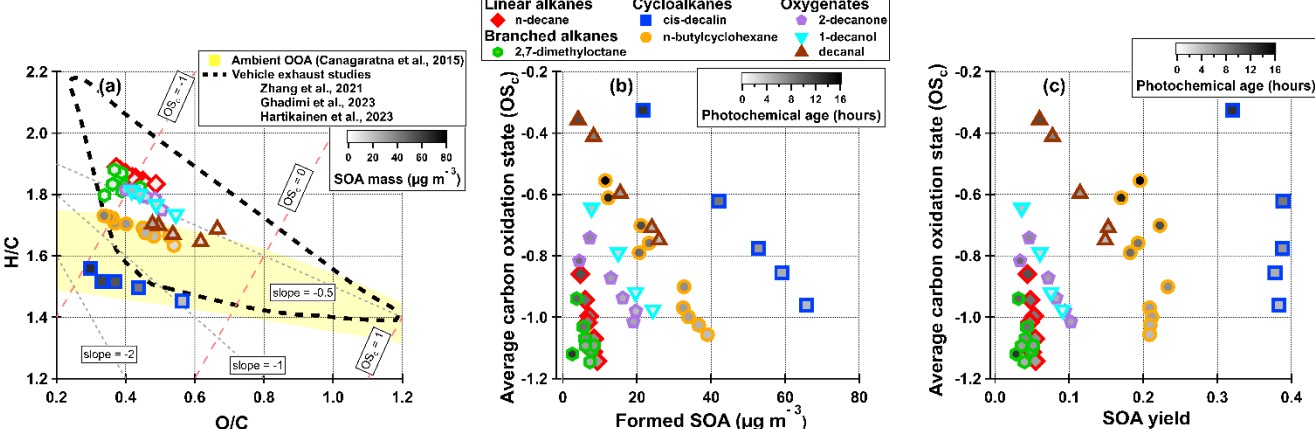

Figure 3. (a) Van Krevelen diagram showing the H/C plotted against O/C. The inner color (white to black) corresponds to the SOA mass concentration of the data point. The red dashed lines correspond to average carbon oxidation states (OS$_c$) and the grey dashed lines are different slopes to guide the reader. The yellow shaded are represents the ambient H/C and O/C range of OOA according to Ng et al. (2011) and improved by Canagaratna et al. (2015). The area within the black dashed line represents data from recent vehicle emissions studies (Zhang et al., 2021; Ghadimi et al., 2023; Hartikainen et al., 2023). Average carbon oxidation state plotted against (b) SOA mass concentration and (c) SOA yield. The inner color (white to black) corresponds to the photochemical age (in hours) of the data point.

The oxygen to carbon (O/C) and hydrogen to carbon (H/C) ratios are calculated according to the improved ambient method described in Canagaratna et al. (2015), and plotted in a Van Krevelen diagram in Fig. 3a. The average carbon oxidation state ($OS_C$) is a parameter that is proposed to describe better the degree of oxidation than pure O/C ratios (Kroll et al., 2011), and can be approximated as $OS_C \approx 2 \times (O/C) - (H/C)$. $OS_C$ is plotted against formed SOA mass concentration in Fig. 3b and SOA yield in Fig. 3c.

In general, for each individual compound, we clearly see that with increasing SOA mass concentration, the O/C ratio and $OS_C$ decreases and the H/C ratio increases (Fig. 3a and b). This is in good agreement with previous studies (Shilling et al., 2009; Kuwata et al., 2012; Day et al., 2022). At low loadings, only the least volatile species, which are generally the most oxidized, are able to condense and form SOA, but as the particle mass increases, more volatile (, less oxidized and shorter photochemical aged) compounds can condense onto the particles, leading to decreased O/C and $OS_C$ and increased H/C.

The H/C ratios of the SOA follows roughly the same order as the H/C ratios of the precursors (ranging from 1.8 to 2.2). Furthermore, our elemental ratios and slopes for cis-decalin and n-decane SOA are similar to those in Li et al. (2019).

Figure 3c reveals an interesting feature of the alkane compounds (linear, branched and cyclic): the SOA yield is relatively stable even if the $OS_C$ increases. This is different to the oxygenated compounds, that exhibits a stronger trend with higher SOA

yields at lower $OS_C$. Especially cyclic compounds are able to maintain high SOA yield regardless of the oxidation state, indicating again that these compounds are efficient on producing low-volatile oxidation products for SOA formation.

Only the high-yield compounds go into the ambient H/C and O/C range (yellow area in Fig. 3a, (Canagaratna et al., 2015)),
while the other compounds have higher H/C ratio than the ambient range. The precursors in our study have quite high H/C ratio, which could explain why they are above the ambient H/C range. Already Ng et al. (2011) observed that hydrocarbon-like OA (HOA) lies outside the ambient range (higher H/C ratios), which can partly explain why our results shows a similar trend; emissions from fossil fuel combustion, consisting of mainly alkanes, can be classified as HOA. Furthermore, all data, except cis-decalin, falls into the area that represents recent vehicle exhaust studies (Zhang et al., 2021; Ghadimi et al., 2023;
Hartikainen et al., 2023). As alkanes are mostly anthropogenic emissions, these studies represent our experiment better than most typical ambient SOA studies, which can include biogenic SOA from precursors with typically much lower H/C ratios.

## 3.2 HOM comparison

We detected HOMs (compounds with six or more O-atoms) from all precursors and some example spectra are shown in Fig. A2-A8. While our own HOM measurements did not allow for yield determinations, we can compare the SOA yields from our study to the HOM yields from Wang et al. (2021). While the group of studied VOCs for our study was chosen from Wang et al. (2021), the experimental conditions are not the same. Wang et al. (2021) used a flow reactor with 3 s residence time and high VOC concentrations (~10 ppm), which also explains why they did not detect any second-generation oxidation while we
did (discussed in more detail below). However, a comparison is shown in Fig. A9, which shows our SOA yields versus the molar HOM yields in Fig. 3 from Wang et al. (2021). Figure A9 shows that decanal and cis-decalin had the highest HOM molar yields, which aligns well with our study as those two compounds belongs to the group of SOA yields compounds. Furthermore, Wang et al. (2021) did not see any HOM signal for n-decane or 2,7-dimethyloctane (or measured it for 2-decanone), whereas we were able to get HOM signal from all of them. For the four VOCs that showed measurable HOM yields
in Wang et al. (2021), are the same VOCs that resulted in the four highest SOA yields in our study (excluding 2-decanone which was studied only in this study). Although the comparison between our study and Wang et al. (2021) is not ideal, this is the first comparison of HOM and SOA yields for alkanes and their oxygenates derivatives, and we can see an indication that higher HOM yields correlates with higher SOA yields.

While our HOM data was not useful for quantification, the HOM distributions can be of use when assessing the role of multi-generational OH oxidation. If we assume, in the absence of $NO_x$, that the generic oxidation pathway for the precursors starts with H-abstraction by OH, and potential radical propagation of peroxy ($RO_2$) or alkoxy (RO) radicals continue until radical termination takes place through the loss of OH or $HO_2$, the formed products will have two H-atoms less than the precursors. If the products undergo another reaction with OH, the second-generation oxidation products would have lost two additional

280 H-atoms. As an example, in the case of cis-decalin ($C_{10}H_{18}$), we would expect first-generation products to consist primarily of $C_{10}H_{16}O_z$ compounds under our experimental conditions, while second-generation products would consist mainly of $C_{10}H_{14}O_z$ compounds. We indeed observed that the ratio of second to first generation products (i.e., $C_{10}H_{14}O_y$ to $C_{10}H_{16}O_y$ (y=4-9)) increased with decreasing injected VOC concentration (Fig. 4), indicating that lower VOC concentration lead to increased second-generation OH oxidation in our system. This is to be expected, as the OH production stays largely constant, and thus

high injected VOC concentrations will lead to OH radicals reacting almost solely with VOC molecules instead of oxidation products. However, at lower injection rates, the VOC concentration will decrease significantly and there will be more OH radicals available to react with oxidation products. Figure 4b shows the same ratios plotted versus the reacted amount of VOC. The markers inner color (white to black), that shows the fraction of reacted VOC. At lower injection rates, a higher fraction of the VOC is reacted, leaving more OH to react with oxidation products, while at higher injection rates, a lower fraction (but a

higher absolute concentration) is reacted. This expected behaviour can be seen for all precursors (Fig. 4). This result, and the overall high values for these ratios (mostly around or above unity) suggest that multi-generation OH oxidation is of importance in our experiments. Although our setup was not optimized for HOM detection, we were able to capture the behaviour of decreasing average H/C ratio with increasing VOC concentration. These findings are similar to studies done with aromatic compounds in Garmash et al. (2020).

For simplicity, as different compound ratios are calculated for the different precursors, we will hereafter call the ratios $H_n:H_{n+2}$ ratio. Interestingly, the cyclic compounds, which also had the highest SOA yields, show the highest value of the $H_n:H_{n+2}$ ratio. Speculating, if the cyclic compounds produce more oxidized products, which then would have higher reactivity against OH, the number of second-generation products would be higher and therefore explain this behaviour. The other compounds have lower values for the $H_n:H_{n+2}$ ratio, and go below unity at higher VOC concentrations. This suggest that for 2,7-dimethyloctane,

n-decane, 2-decanone and decanal second- (or multi-) generation oxidation is not as important as for cis-decalin, n-butylcyclohexane and 1-decanol under these conditions.

As discussed in previous sections regarding the SOA formation, our measurements were done in the absence of $NO_x$, which can change, in addition to SOA yields, the gas phase products and reactions pathways. However, similar to $RO_2 + NO$ reactions, the main products of $RO_2 + RO_2$ reactions are typically also RO radicals, and thus RO-initiated autoxidation can occur in a

305 very similar way in both systems. Differences in exact yields of RO radicals will vary between the two reaction pathways, but will also vary for the same pathways depending on the structure of R. The yields of organic nitrates from $RO_2 + NO$ can vary strongly, while yields of accretion or alcohol/ketone products can similarly vary for $RO_2 + RO_2$.

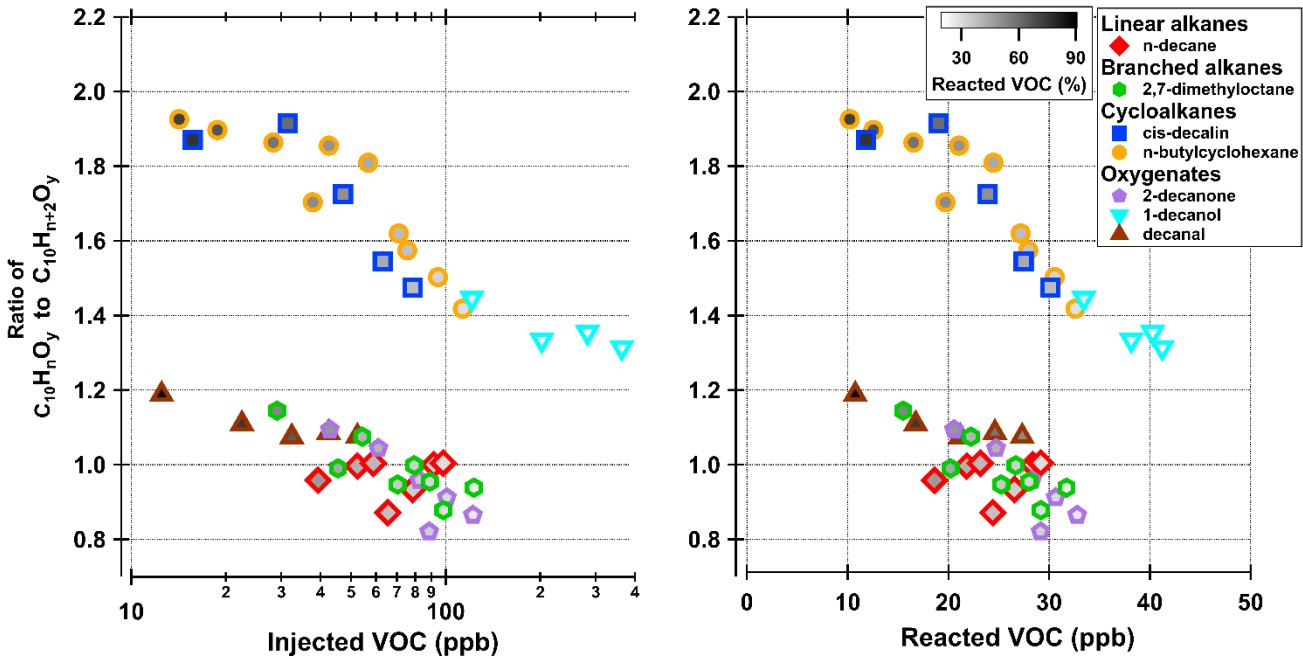

**Figure 4. Ratios indicating the importance of multi-generation OH oxidation. The ratios on the y-axis are presented as $C_{10}H_nO_y/C_{10}H_{n+2}O_y$, with y=4-9 and n=14 for cis-decalin, n=16 for butyl-cyclohexane, 2-decanone and decanal, and n=18 for n-decane, 2,7-dimethyloctane and 1-decanol. The inner color (white to black) corresponds to the reacted VOC (%) of the data point. For visualization, the colorscale range is set from 20 % to 90 %, which includes all precursors except 1-decanol that ranges from 11 % to 28 %.**

## 4 Conclusions

We conducted experiments using a PAM chamber to measure SOA yields for seven $C_{10}$ alkanes and their oxygenated derivatives. The SOA formation was initiated by OH oxidation, in the absence of $NO_x$ and seed aerosol. The conditions in the PAM chamber were equivalent to approximately 1 to 16 hours of atmospheric aging, producing mass concentrations typically found in the atmosphere ($< 70 \mu g \ m^{-3}$). From our subset of $C_{10}$ VOC, cis-decalin, decanal and n-butylcyclohexane showed the highest SOA yields. For alkanes, the SOA yield increased in the order of cyclic > linear > branched alkanes, in accordance with earlier observations. We also compared our SOA yields to previous reports of yields of highly oxygenated organic molecules (HOM)(Wang et al., 2021), finding that higher HOM yields indicates higher SOA yields.

In addition, we found clear differences in the contributions of multi-generational OH oxidation for the different precursors. This finding was based on assessing the amounts of H-atoms in the observed gas phase oxidation products that were measured from the PAM. In general, lower precursor concentrations enhanced multi-generational OH oxidation.

The primary aim of this study was to assess SOA yields of the $C_{10}$ VOCs and compare them to observed HOM yields. Our findings suggest a link between the two yields, with compounds having low HOM yields also having low SOA yields. However, the HOM and SOA yields were determined from different experiments under different conditions, meaning that

quantification of the role of HOMs for the SOA formation is not possible from this work. In particular, the role of multigeneration OH oxidation was clearly higher in our study than in the one where HOM yields were determined, and even in our experiments there was considerable variation in the role of multigeneration oxidation between the VOCs. Overall,

determining the exact chemical mechanism forming SOA from alkanes and their oxygenates will require further investigations, but our findings indicate that autoxidation and HOM formation should be accounted for when performing such investigations.

## Appendix A: Supporting figures

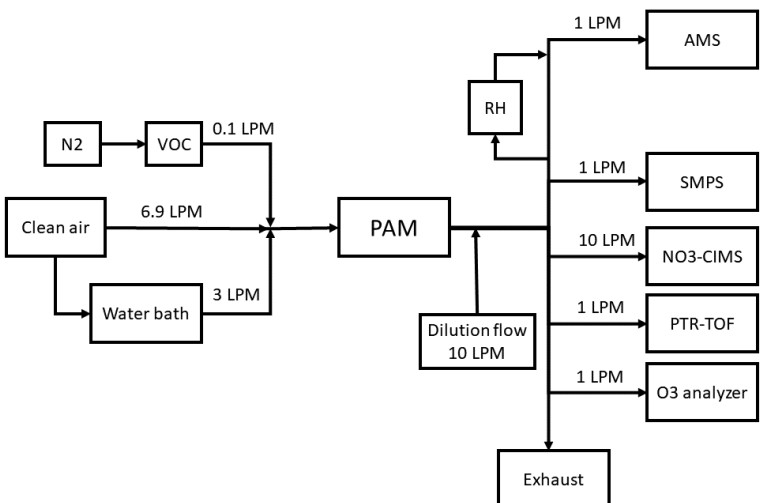

**Figure A**1**. Experimental setup used in this work. Total flow to the Potential Aerosol Mass (PAM) chamber consisted of (1) a nitrogen**

**(N₂) flow where single VOC (volatile organic compound) was injected, (2) humidification flow (through a water bath) and (3) a clean air flow. PAM outflow was diluted with a dilution flow. Particle phase products were measured with a Long Time of Flight Aerosol Mass Spectrometer (AMS) and a Scanning Mobility Particle Sizer (SMPS). Gas phase products were monitored with a nitrate chemical ionisation MS (NO₃-CIMS) and a Proton Transfer Reaction ToF MS (PTR-TOF) and an ozone (O₃) analyzer.**

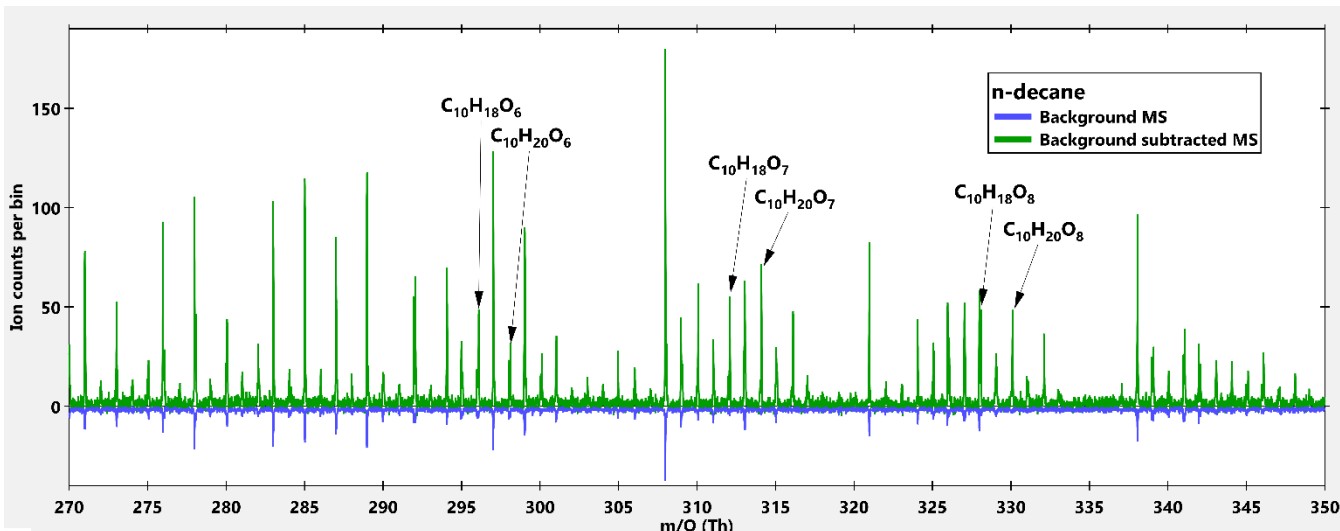

Figure A2. High resolution mass spectra (MS) of n-decane + OH. Blue MS is the background while the green MS is the background subtracted MS. The background MS is multiplied by a factor of -1 for visualization reasons.

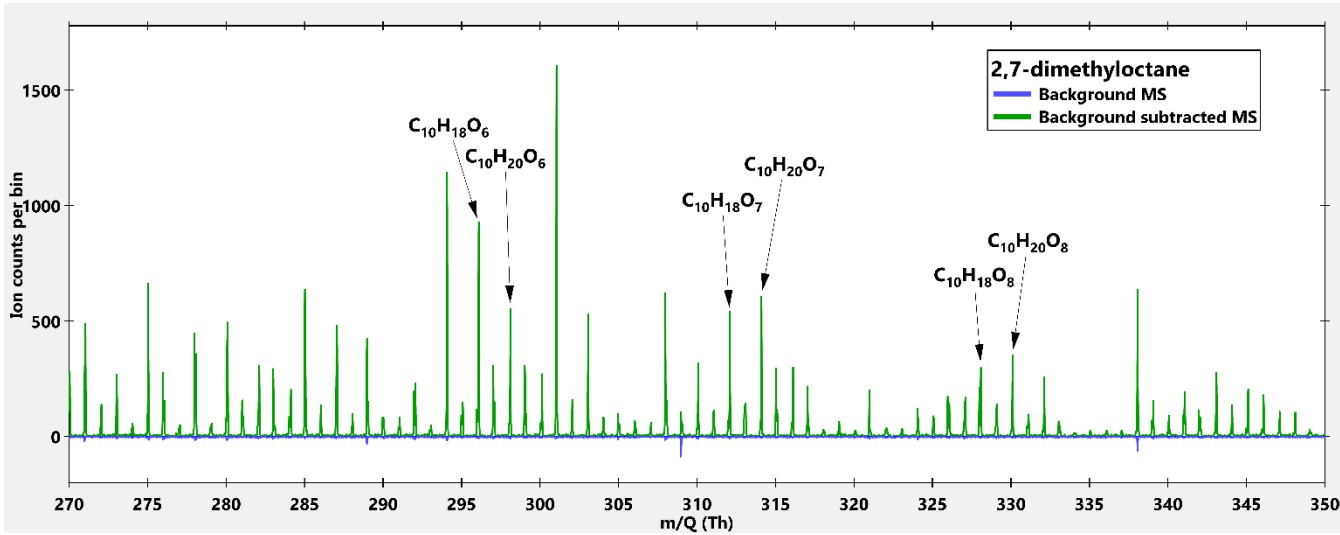

Figure A3. High resolution mass spectra (MS) of 2,7-dimethyloctane + OH. Blue MS is the background while the green MS is the background subtracted MS. The background MS is multiplied by a factor of -1 for visualization reasons.

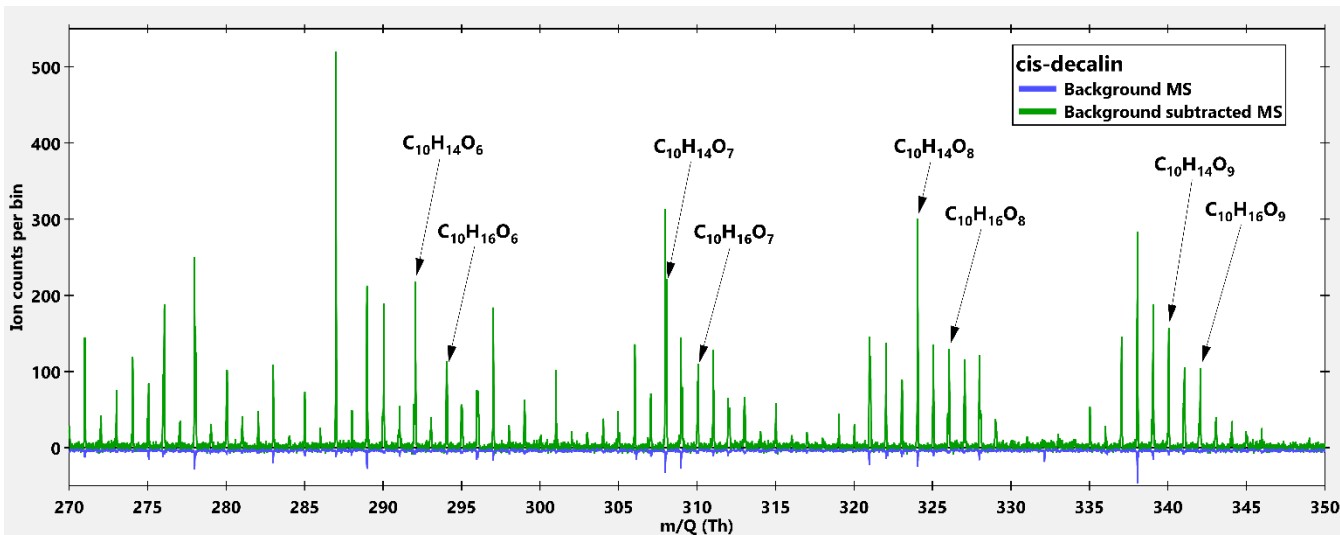

**Figure A4.** High resolution mass spectra (MS) of cis-decalin + OH. Blue MS is the background while the green MS is the background subtracted MS. The background MS is multiplied by a factor of -1 for visualization reasons.

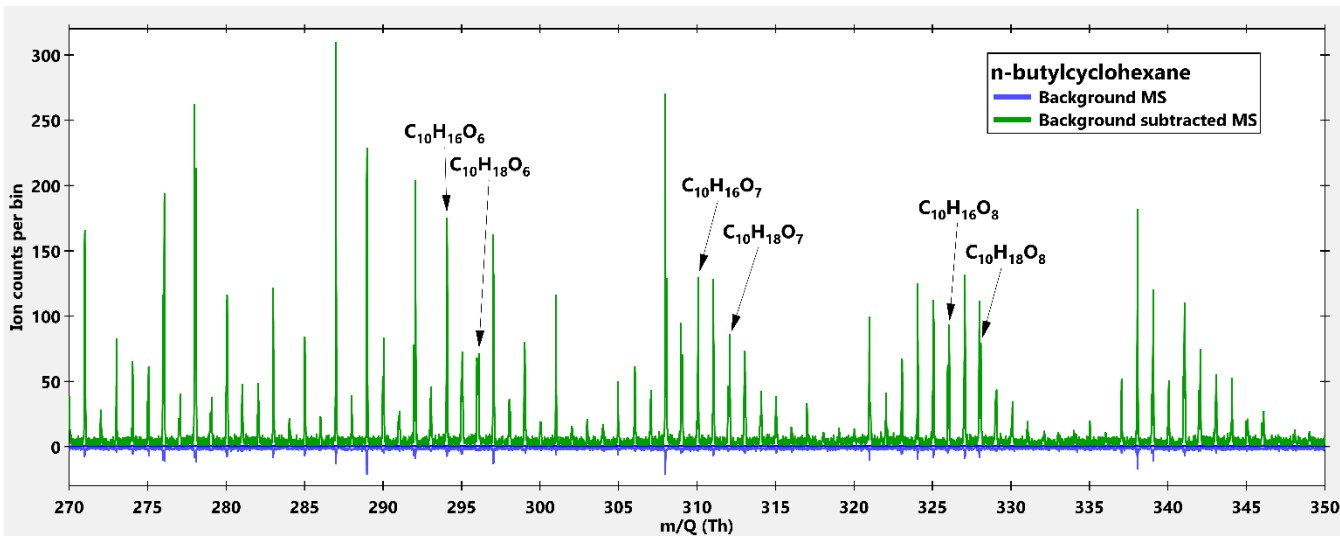

**Figure A5.** High resolution mass spectra (MS) of n-butylcyclohexane + OH. Blue MS is the background while the green MS is the background subtracted MS. The background MS is multiplied by a factor of -1 for visualization reasons.

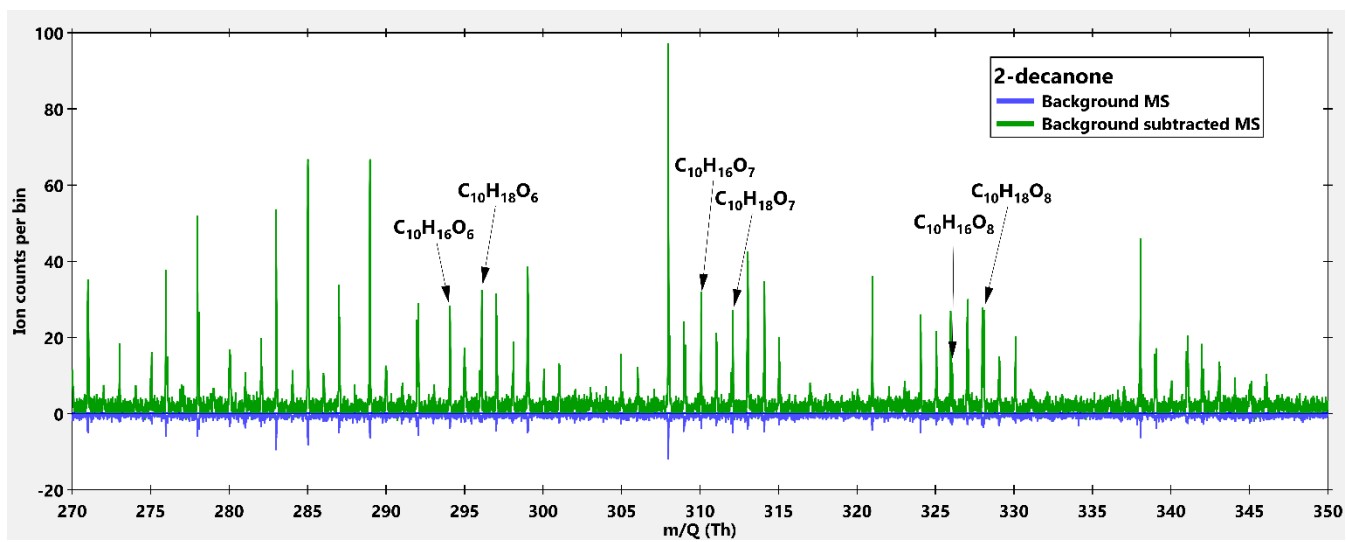

**Figure A6. High resolution mass spectra (MS) of 2-decanone + OH. Blue MS is the background while the green MS is the background subtracted MS. The background MS is multiplied by a factor of -1 for visualization reasons**.

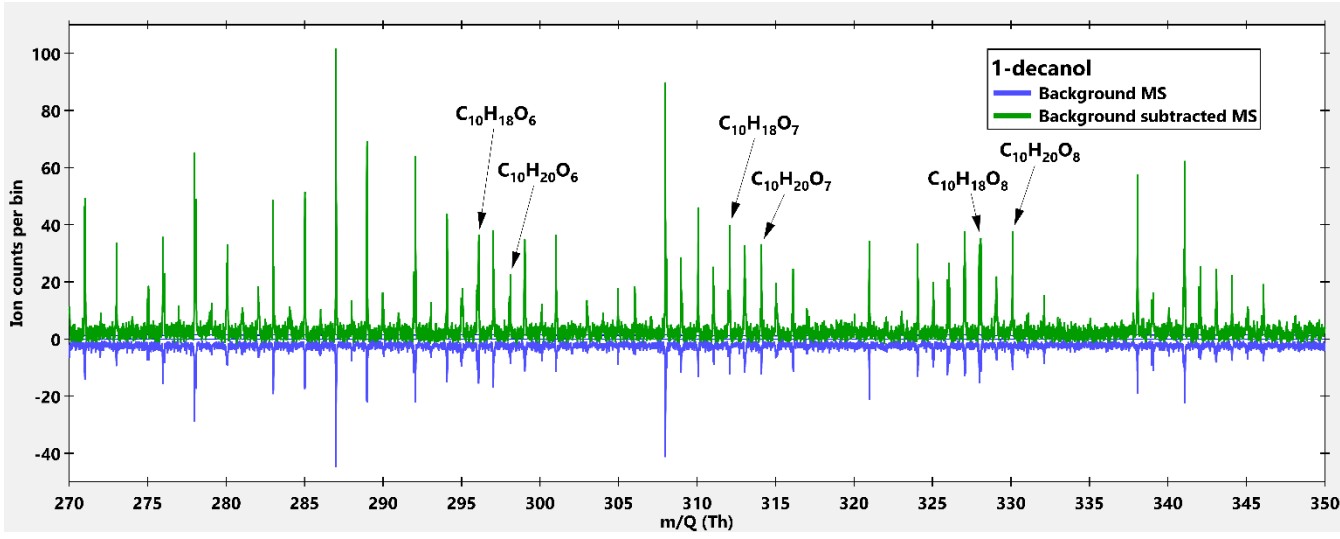

**Figure A7. High resolution mass spectra (MS) of 1-decanol + OH. Blue MS is the background while the green MS is the background subtracted MS. The background MS is multiplied by a factor of -1 for visualization reasons.**

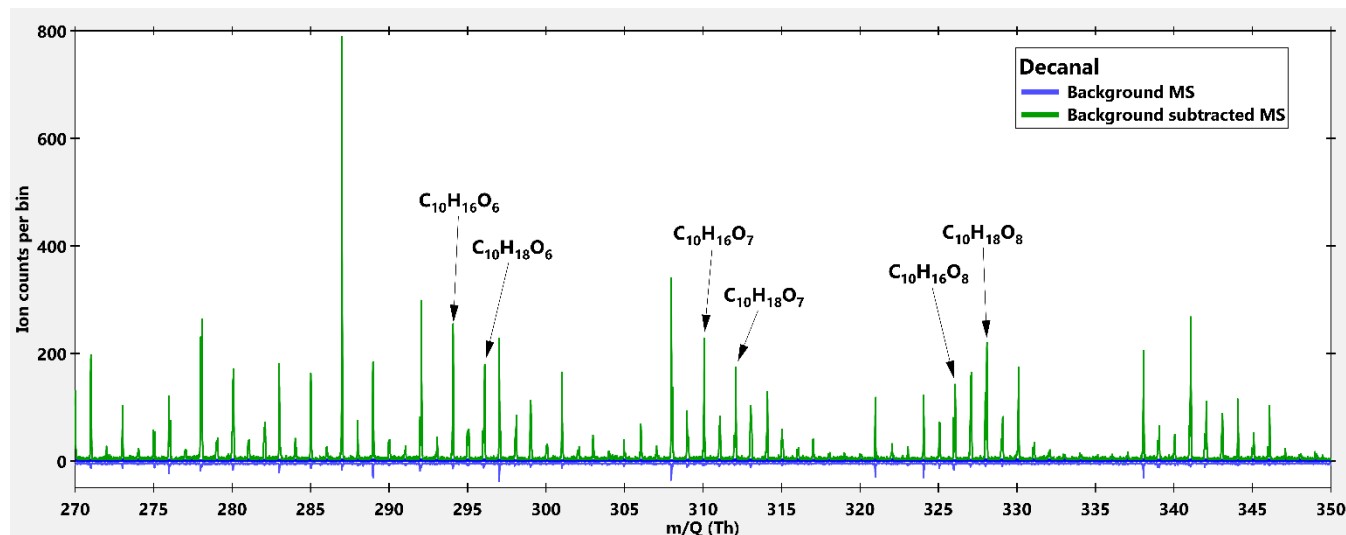

**Figure A8. High resolution mass spectra (MS) of decanal + OH. Blue MS is the background while the green MS is the background subtracted MS. The background MS is multiplied by a factor of -1 for visualization reasons**.

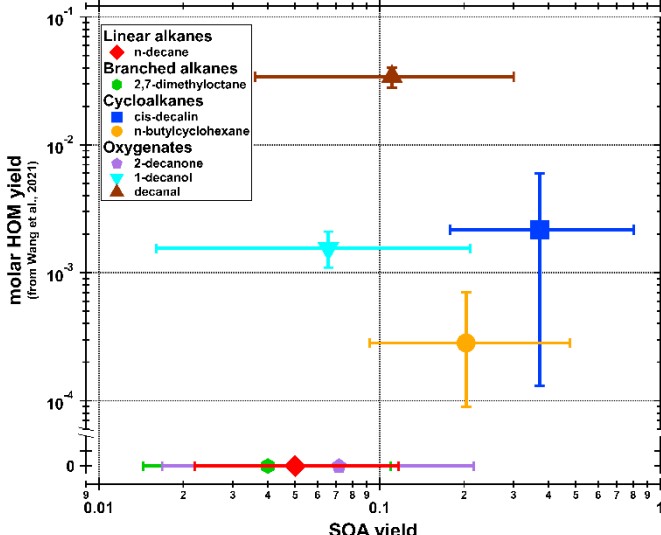

**Figure A9. SOA yields (from Fig. 2) versus HOM yields (from Fig. 3 in Wang et al. (2021)). Datapoints are the average and the error bars represents the minimum and maximum of the measurements. For 2,7-dimethyloctane and n-decane, HOMs were not detected by Wang et al. (2021), and 2-decanone were not included in their HOM measurements. Therefore, the HOM yields for these three compounds are assumed to be zero.**

## Appendix B: Corrections and uncertainties for calculating the SOA yields

As described in Section. 3.1.1., the SOA yields (Y) are calculated as the ratio of formed organic aerosol concentration ($C_{OA}$) to reacted precursor concentration ($\Delta VOC$) by:

$$Y = C_{OA}/\Delta VOC \qquad (1)$$

, where $\Delta VOC$ is:

$$\Delta VOC = [VOC] \times \left(1 - e^{-k \times OH_{exp}}\right) \qquad (2)$$

, where k is the second-order rate constant of the precursor with OH and [VOC] is the injected VOC.

The following uncertainties has been applied in the error calculations:

- initial VOC concentration: ± 5 %
- PAM residence time: ± 10 %
- reaction rate constant: ± 15 %
- $OH_{exp}$: ± 20 %
- SOA measurements: ± 30 %

Initial VOC concentration is crucial for calculating SOA yields, and it was calculated from the injected volume by the syringe pump. Unfortunately, the PTR-ToF was neither functioning throughout the measurements or successfully calibrated, so we cannot compare any measured VOC concentrations with the calculated one. However, the syringe pump was calibrated prior the measurements, and the injected volume can be trusted. Despite that, the add an uncertainty of ± 5 % to the initial VOC concentration.

The PAM residence time is needed in the $OH_{exp}$ model by Li et al. (2016), where we assumed ideal plug flow when calculating the residence time. However, in e.g., Lambe et al. (2011) the actual measured residence time distribution (RTD) can differ significantly from the ideal plug flow. As we did not perform any measurement of the RTD for our setup, we assume that our RTD is similar to the one in Lambe et al. (2011). By testing how the SOA yield changes with different modelled $OH_{exp}$ (by increasing and decreasing the residence time), we noticed there is only small changes in SOA yield (< 5 %) even if the residence time would be ± 20 %. Therefore, we conclude that our calculated residence time by assuming ideal plug flow is a good proxy for our setup. However, we still add an extra uncertainty of ± 10 % for the final error calculations.

Reaction rate coefficients are needed in both the modelled $OH_{exp}$ and the $\Delta VOC$ calculations. However, there is always some uncertainty in these measured values (Atkinson, 2003). Therefore, we have added an extra uncertainty of ± 15 % to account for this.

Additional uncertainty comes with modelled $OH_{exp}$ as we do not have calibration measurements over the whole range of conditions used in this study. While the measured $OH_{exp}$ from the nonanal calibration is higher than the modelled $OH_{exp}$ for the same conditions, we decided to use the modelled $OH_{exp}$ for all our calculations as it can capture the changes in different condition for the different SOA precursors. All of the abovementioned parameters (initial VOC

concentration, PAM residence time and rate constants) are already included in the $OH_{exp}$ model. However, we will add an additional uncertainty of $\pm$ 20 % to account for any other uncertainties arising from the modelled $OH_{exp}$.

To account for the AMS and SMPS derived SOA mass, we use an uncertainty of $\pm$ 30 %. This includes the uncertainty of both particle phase instruments, as well as the uncertainty from the LVOC fate model (described in Section 2.1).

When calculating the SOA yield errors in Fig. 2, we calculated separately the upper and lower error values. This was done by altering all the variables within their uncertainty ranges in the SOA yield function to find the minimum and maximum values. This method will not result in symmetrical upper and lower errors, as seen in Fig. 2.

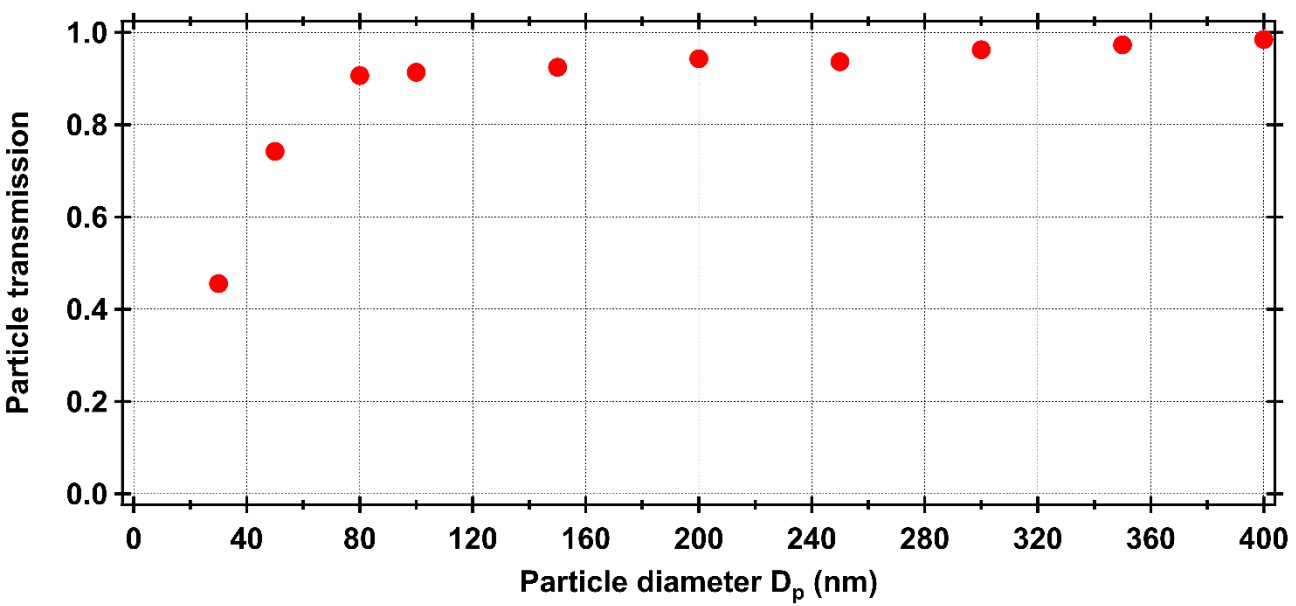


**Figure B1. Particle transmission of our PAM setup as a function of particle size**

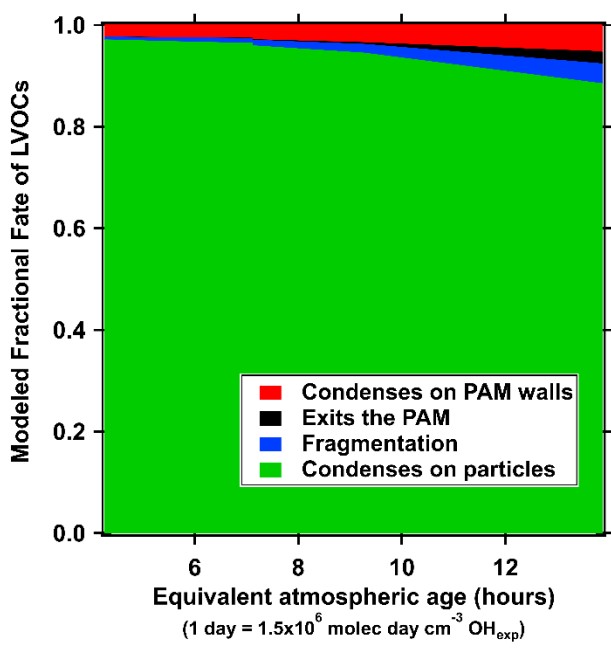

**Figure B2. Fraction of different fates for LVOCs in the PAM-chamber for one of the n-butylcyclohexane experiments. Model from Palm et al. (2016) is used to produce the figure.**


**Data availability.** Data are available upon request by contacting the corresponding author.

**Author Contributions.** ME designed the study. FG led the experiments with the help of KK. FG analysed the data. FG wrote the original draft. HT provided the PAM and support for its usage. All authors commented on the manuscript.

**Competing interests.** The authors declare that they have no conflict of interest.

**Acknowledgments.** This work was supported by the Academy of Finland (grants 320094, 317380 and 345982) and the Jane and Aatos Erkko foundation. Frans Graeffe thanks Svenska Kulturfonden for their support (grants 167344, 177923 and 188272).

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
