# Peer review of "SOA yields from C10 alkanes and oxygenates"

_EGUsphere, 2025_

## Referee Comment (RC2)

**Review Report Graeffe et al. 2025: SOA yields from C10 alkanes and oxygenates and their relation to highly oxygenated organic molecules (HOM)**

Graeffe et al. present a study about the Secondary Organic Aerosol yield from a number of C10 compounds from photooxidation in the presence of ozone and under very low NOx concentrations in an Oxidative Flow Reactor (OFR). The list includes n-alkanes, branched, mono and bicyclic structures as well as the oxygenated compounds (aldehyde, ketone, alcohol). They find the expected trends for the SOA yields and try linking them with the observed Highly oxygenated Organic Molecules (HOM).

Such systematic, fundamental work furthers our understanding of SOA formation mechanisms in the atmosphere and is thus of interest for the audience of this journal. However, I found one important weakness in their methodology (the estimation of the VOC concentration, see Major Comment #1). Before publication, the impact of this issue needs to be discussed in detail to fully evaluate the findings of this study.

**Major comments**

1) Knowing the amount of reacted VOCs is crucial for the calculation of aerosol yields. Apparently due to some instrumental misfortune, the VOC concentrations (ingoing and outgoing) could not be measured for most of the experiments. The authors rely on the calculated concentrations from the injection method (flow of syringe pump) for the initial VOC concentration in the OFR. They then calculate the reacted amount from the OH exposure and known reaction constants. This method in itself is a valid approach if no measurements are available. But there are several sources of uncertainty for the calculated reacted VOC concentration which is the crucial parameter for the yield calculations.

   a. Uncertainty of the initial VOC concentration. Do the authors have any indication of the accuracy of this estimation? E.g. how do the calculated and measured values of the initial VOC concertation compare for a case where they do have PTR data? (e.g. the OH exposure experiment with nonanal)

   b. OH exposure. The authors use the OH estimator model from Li et al. (2016). They compared the calculated value with a test measurement for nonanal (a compound not used in the actual yield experiments) and report a more than two times higher OH exposure value from the measurements than from the model. Doing a simple calculation using a k(OH) of 1e-11 cm3 molec-1 s-1, I calculate that the higher OH exposure leads to a 1.8 times higher amount of consumed VOC (see table below). The yield would be affected by the same factor.

   *Table 1: Calculation of consumed VOC with measured and modelled OH exposure*

| | | |
|---|---|---|
| OHexp(meas) | 6.20E+10 | molec. /cm3/s |
| OHexp(model) | 2.90E+10 | |
| | | |
| VOC initial | 35 | ppb |
| k | 1.00E-11 | cm3/molec./s |
| | | |
| VOC | consumed | |
| meas | 16.2 | ppb |
| calc | 8.8 | ppb |
| | | |
| meas/calc | 1.835474 | |

Since the authors use the model for all experiments, potentially all VOCconsumed values are too low. Then the calculated yield would be too high. But this is just my first "top-of the-head" thought and the authors need to go into the details.

c. Residence time. It was not said explicitly, but I assume that the stated residence time was derived assuming plugged flow. PAM characterisation papers showed that due to the inlet geometry, the flow inside PAM is not represented by the "plugged flow" simplification. Fig 3 in (Lambe et al., 2011) shows an example of how much plugged flow and the actual residence time distribution can differ. The residence time is not used directly in the calculation, but it is folded into the calculated OH exposure where one single value is used for the residence time. Do the authors have any measurements for the residence time distribution in their PAM with their specific flow setup? What would the plugged flow residence time represent? Do most VOC molecules experience a shorter residence time? Is the plugged flow a representation of the average residence time? How would this affect the calculated vs measured OH exposure and thus ΔVOC?

d. Reaction constants. What values were used for the reaction coefficients? I did not see any values or reference. Like any other parameter, reaction coefficients will have an uncertainty which will directly contribute to the uncertainty of the calculated ΔVOC concentration. From the typical values found in literature for these type of compounds (e.g. (Shaw et al., 2020)), I would guess a 10-20% uncertainty in the reaction coefficient values could be reasonable. How much does that add to the overall uncertainty for ΔVOC and thus the uncertainty of the aerosol yields?

With all this in mind, the authors should provide a thorough discussion of the overall uncertainty of their estimated VOC concentration values and how that will affect the calculated SOA yields. E.g., if the reported values are a lower or higher estimate and how much additional uncertain stems from the VOC estimation. Such uncertainty needs to be taken into account in the discussion when comparing the yield values from this study with literature values.

2) The HOM part is too superficial in my opinion. I understand that the experiment design was not favourable for quantitative HOM measurements. But only two C10 HOM groups seem to be investigated in the text. In the appendix figures A2-A8, 6-8 ions (all C10) are identified. These do not seem to be the most prominent (strongest) signals in that m/z range. Why were these chosen? Are these identified HOM representative for the overall HOM population? Does the molar HOM yield in Fig 4 include all potential HOM in the system or also only a specific subset (e.g., only C10 compounds)? (See also specific comment 24)
In my opinion, the authors could strengthen the scientific impact of this manuscript by expanding the discussion of the HOM compounds.

**Specific comments**

1) Line 48: what is the "bimolecular reaction rate" referring to? O2 addition to the alkyl radical? Or the initial OH+VOC reaction?
2) Lines 72-74: The description of the oxidant formation is not strictly precises. OH is not produced from water vapour, but from photolysis of O3 and then consecutive reaction of O(1D) with H2O. HO2 radicals are mostly formed from photolysis but from the reaction of OH with O3 (or VOCs). HO2 is listed as an "oxidant" in the same way as OH is. But to my knowledge. HO2 will not initiate oxidation reactions with the investigated VOCs but rather participate in the reaction mechanism, e.g. terminating RO2 radical chemistry. This sentence needs rephrasing and clarification.

3) Line 76 & Fig A1: To my knowledge, most PAM systems use a Nafion humidifier and not a water bath for humidification.

4) Line 76: 22% RH is a rather low value. Often 40% RH are used in OFR or chamber studies as a representation of average atmospheric conditions. Furthermore, the recommendation to obtain atmospherically comparable VOC and RO2 chemistry for OFR185 mode is to use high H2O, low UV, and low OHRext (see conclusions of (Peng et al., 2019)). Was there a reason for choosing 22% or did this stem from the limitation of the setup (i.e., the efficiency of the humidifier at the high flow needed for the instrumentation)?

5) Line 79: The UV lamps were set to 100V? To my knowledge PAM UV lights use a control voltage of 0-10 V. Was a different system used? Or is this a different voltage?

6) Line 77: I assume the residence time value is calculated assuming plugged flow. This needs to be clarified.

7) From the description, I assume PAM was operated in "OFR185" mode, i.e., using the 185 nm UV lamps inside PAM to generate O3 and OH at the same time? Using this label may help clarify the mode of operation and facilitate comparison with other studies.

8) Line 111 and Figure A1: I understand that additional dilution after PAM was necessary to achieve enough sample flow. The sampling flows in Fig A1 add up to 14 lpm. Why was the dilution flow set to 10 lpm creating an overflow of 6 lpm?

9) Line 89f: I do not understand how the authors come to the conclusion that the measurements do not include the external reactivity. The presence of nonanal in the measurements is the "external OH reactivity". The derived OH exposure values will be the values in the presence of the set amount of nonanal.
I assume that the authors used the "OFR exposure estimator" based on Li et al. (https://sites.google.com/site/pamwiki/estimation-equations). Then the reason that the model estimates a lower value is that it assumes that the set external reactivity is present for the whole residence time. This is equivalent with assuming that the 35 ppb of nonanal are never consumed. Or that the reaction products of nonanal continue to react with OH at the same rate as the precursor and that they continue to do so for the whole residence time. Neither of these are of course correct – the truth is somewhat in the middle (precursor gets consumed, reaction products react). However, the discrepancy between the model and measurements does not stem from "not include the external reactivity".

10) Line 96: If the authors provide the information about the AMS data acquisition interval, they should also mention if they were only alternating between open and closed or if they also ran pToF. Typical settings are 20sec for open/closed. Then only 2/3 of the 1 min would actually contain MS data.

11) Line 97: what is meant with an "overflow of 1L/min"

12) Line 102: what is meant by "area concentration"?

13) Line 104: How much uncertainty is introduced by using this density estimation? If AMS did acquire pToF data, do the estimated densities match the densities that can be derived from comparing aerodynamic and electromobility diameters (at least the trends between experiments)?

14) Line 102: The upper limit of the SMPS was 500nm (electromobility). How did the volume size distribution spectra look? Was there considerable mass/volume at the highest size and could this be an indication that some particles >500nm were omitted in the SMPS measurements? Few particles at that size can already contribute a large portion of the aerosol mass.

15) There is no mention of any O3 scrubber being used. Was this indeed the case?

16) Line 155ff: I do not agree with the interpretation of SOA volatility from the yield data in this way. To me the point is that decaline produces a larger fraction of low volatility material already at low VOC conc. The acyclic ones produce less of those compounds. Hence, less SOA is formed. But the volatility of the condensing products cannot be derived from this. E.g., let's assume that the system allows particle phase partitioning of compounds with $C^* < 1e-4$ ug/m3. VOC 1 has 10% of its oxidation products with $C^* = 1e-4$ ug/m3 and nothing below that. VOC 2 has 1% products at $C^* = 1e-6$ ug/m3. VOC 1 will form more SOA than VOC 2. At low precursor concentrations, VOC 2 may not

form any SOA as it does not create high enough concentrations to initiate nucleation (wall losses may also play a role). At higher precursor concentrations, SOA is formed in both cases. But the volatility of the SOA would be higher for VOC 1 then for VOC 2 – so the opposite of what would be expected from the SOA yield.

A real example for a precursor with lower aerosol yield and also lower SOA volatility is the comparison of SOA from farnesene and a-pinene in (Ylisirniö et al., 2019).

My point here, aerosol yields cannot be interpreted directly into SOA volatility in this way without further composition information or information about the volatility distribution of the gas and particle phase products.

17) Section 3.1.2: was the basic parametrisation (Aiken et al., 2007) or the "improved" parametrisation (Canagaratna et al., 2015) used to derive O:H and H:C values from AMS? This should be stated in the methods section.

18) Line 191ff: The trends of O:C with overall yield seem to suggest that higher yields are linked to higher O:C values. But the decanal points are all >0.5 while decaline and butylcyclohexane shows O:C values that go much lower. But all decaline and butylcyclohexane points have a higher yield than even the highest decanal points. With this in mind, can the authors really make this claim about the trends?

19) Section 3.1.2: Both H:C and O:C values vary for all precursors. Did the authors look into using OSc (average oxidation state of carbon), a parameter which combine O:C and H:C, instead of just individually O:C and H:C? Would that reveal clearer trends?

20) Line 210ff: if the conditions were so different (namely VOC to oxidant ratio and residence time), are the HOM yields representative for this study?

21) Generally, HO2 concentrations are very high in PAM when operated at low NOx (very little recycling back to OH). How would that affect HOM production. Don't higher HO2 concentrations enhance the quenching of RO and RO2 radicals and thus suppress auto-oxidation type processes?

22) Fig 4: I wonder how useful this figure/comparison is knowing about the limitations of the HOM data. It may be doing more harm than good. Playing devil's advocate, I can look at this figure and state that decanal can show a aerosol yield of 0.08 with molar HOM yield of ~3e-2 and decanone can have the same aerosol yield with "no HOM at all". Thus, HOM cannot be that important for aerosol yields Looking only at the precursors that had a measurable HOM yield, I could claim that and increasing HOM yields show decreasing aerosol yields (decanal& decanol vs decaline and butylcyclohexane).

23) Line 237:"and thus high injected VOC concentrations will lead to the majority of OH radicals reacting with the VOC. " I'm not sure if this is strictly true for PAM. A lot of the OH radicals will also react with O3 (which is at 10s of ppm I assume).

24) Line 230ff: Only C10 HOM compounds were investigated for the evaluation of the first/second generation products. Does this interpretation hold when other HOM species are included (e.g. with C9 or C8)?

25) Line 230ff: Does C-C bond cleavage become more important on successive oxidation? If that is the case, would that not mean that some of the second generation product are no longer C10 and thus "hide" as C<10 HOM? Could this be more pronounced for acyclic compounds?

26) Line 230ff: This analysis (first/second generation) is only based on the gas phase data. If the multi gen products of decanal etc. are just a bit more low-volatility, they could be condensing into the particle phase, hiding from the gas phase measurements. Then it would look like there is less of them in the gas phase, right?

27) Line 273ff: I do not agree with finding "a clear link between the two yields" (see specific comment 22). If anything, the combination of the Wang et al. and this new study shows how much the experimental setup and chosen reaction conditions can impact the formed and detected HOM amounts and types. Thus, great care must be taken in the experiment design and when comparing HOM data from different studies.

28) Fig A1: This figure needs more explanations in the caption. None of the acronyms/abbreviations are explained. The main text does contain most of the information, but this figure will become much easier to understand if the information is also provided directly with it.

**Language**

+ line 11: "emitted in the atmosphere" – should be "emitted into"

+ line 112 "got enough of sample flow" – I would omit the "of"

+ line 212 "chosen from them" who is "them"? I guess the auhors mean the Wang paper? Please rephrase to make clearer what the "them" is referring to.

+ line 235f: The way the increase/decrease trends are assigned in this sentence was a bit hard to wrap my head around. To paraphrase: "decrease of A to B ratio with increasing VOC conc means that lower VOC conc increases second gen oxidation" Consider rephrasing this. "the ratio of second to first generation product ions increased with decreasing VOC conc, i.e., lower precursor conc  "

+ line 237: "OH generation" maybe better use "OH production" to differentiate from the other meaning of the word generation (reaction generation) which is used in the same paragraph.

**References**

Aiken, A. C., DeCarlo, P. F., Jimenez, J. L., and L., J. J.: Elemental analysis of organic species with electron ionization high-resolution mass spectrometry., Anal. Chem., 79, 8350–8, https://doi.org/10.1021/ac071150w, 2007.

Canagaratna, M. R., Jimenez, J. L., Kroll, J. H., Chen, Q., Kessler, S. H., Massoli, P., Hildebrandt Ruiz, L., Fortner, E., Williams, L. R., Wilson, K. R., Surratt, J. D., Donahue, N. M., Jayne, J. T., and Worsnop, D. R.: Elemental ratio measurements of organic compounds using aerosol mass spectrometry: Characterization, improved calibration, and implications, Atmos. Chem. Phys., 15, 253–272, https://doi.org/10.5194/acp-15-253-2015, 2015.

Lambe, A. T., Ahern, A. T., Williams, L. R., Slowik, J. G., Wong, J. P. S. S., Abbatt, J. P. D. D., Brune, W. H., Ng, N. L., Wright, J. P., Croasdale, D. R., Worsnop, D. R., Davidovits, P., and Onasch, T. B.: Characterization of aerosol photooxidation flow reactors: heterogeneous oxidation, secondary organic aerosol formation and cloud condensation nuclei activity measurements, Atmos. Meas. Tech., 4, 445–461, https://doi.org/10.5194/amt-4-445-2011, 2011.

Peng, Z., Lee-Taylor, J., Orlando, J. J., Tyndall, G. S., and Jimenez, J. L.: Organic peroxy radical chemistry in oxidation flow reactors and environmental chambers and their atmospheric relevance, Atmos. Chem. Phys., 19, 813–834, https://doi.org/10.5194/acp-19-813-2019, 2019.

Shaw, J. T., Rickard, A. R., Newland, M. J., and Dillon, T. J.: Rate coefficients for reactions of OH with aromatic and aliphatic volatile organic compounds determined by the multivariate relative rate technique, Atmos. Chem. Phys, 20, 9725–9736, https://doi.org/10.5194/acp-20-9725-2020, 2020.

Ylisirniö, A., Buchholz, A., Mohr, C., Li, Z., Barreira, L., Lambe, A., Faiola, C., Kari, E., Yli-Juuti, T., Nizkorodov, S., Worsnop, D., Virtanen, A., and Schobesberger, S.: Composition and volatility of SOA formed from oxidation of real tree emissions compared to single VOC-systems, Atmos. Chem. Phys. Discuss., 1–29, https://doi.org/10.5194/acp-2019-939, 2019.

---

## Author Response (AR1)

We thank both reviewers for their insightful feedback. Below we address the comments point-by-point. The original comments are in black, and our responses are given in red and modifications in the revised manuscript are given in *blue*.

**Reviewer: 1**

1.  In this paper, the SOA was produced from the photooxidation of various alkanes in the absence of NOx. As authors described, alkanes are a major part of anthropogenic VOC emissions. This means that the atmospheric oxidation of alkanes is more liked processed in the presence of NOx. In particular, urban air environments are typically in the high NOx region. Authors need to discuss about the potential impact of NOx on HOM productions and alkanes SOA yields.

    Response: The reviewer is correct that a large fraction of alkane emissions will be in environments with high NOx concentrations. We had added more discussion into the manuscript on the potential impacts of NOx, as well as extended the discussion on why we did our experiments in the absence of NOx .
    Although alkanes, and other AVOCs, are mainly emitted in high-NOx environments, they can be transported to low-NOx areas (e.g., the boreal forest), as already mentioned as a motivation for low-NOx studies in Li et al. (2019). In addition, as mentioned already in the manuscript, alkanes are also emitted from volatile chemical products (VCPs, e.g., sanitizers and adhesives) that are often used indoors (low-NOx environment). Therefore, it is also necessary to measure SOA yields at low-NOx conditions.
    Furthermore, as shown and described in Wang et al. (2021), the yields of oxidation products increased (cis-decalin and n-butylcyclohexane) or remained high (decanal and 1-decanol) with increased NO concentration. This is opposite to many BVOCs, where NO addition can suppress HOM formation (and therefore SOA formation). Hallward-Driemeier et al. (2024) showed that five C10 alkanes and oxygenated derivatives exhibits in general higher SOA yields under low-NOx conditions. Therefore, low-NOx conditions can be seen as an upper-limit of alkane SOA yields.

    Therefore, we are confident that our SOA yields are representative although they are measured in absence of NOx. Obviously, the formation of gas- and particle phase organic nitrates alter the chemical composition in the presence of NOx, but the there should not be any rapid decrease or increase in the SOA yields if NOx is added.

    Lastly, as described in Wennberg (2024), low-$NO_x$ vs high-$NO_x$ is not always the best way to describe different atmospheric conditions.

    We have now addressed this in the introductory part of the revised manuscript as:

    *Line 55-65: "The experiments were done in an oxidation flow reactor, in the absence of $NO_x$ and seed particles, simulating fresh SOA. Although alkanes, and other AVOCs, are mainly emitted in the presence of $NO_x$, they can be transported downwind to low-$NO_x$ environments, motivating our study similarly as in Li et al. (2019). In addition, VCPs (including sanitizers and adhesives) are often used indoors (low-$NO_x$ conditions). Furthermore, Hallward-Driemeier et al. (2024) showed that five $C_{10}$ alkanes and oxygenated derivatives exhibits in general higher SOA yields at lower $NO_x$ concentrations, indicating that low-$NO_x$ conditions can be seen as an upper-limit of alkane-SOA yields. Lastly, the comparison of only low-$NO_x$ versus high-$NO_x$ conditions is not strictly simple (Wennberg, 2024), and measuring SOA yields at different $NO_x$ concentrations would be out of scope for this study. Therefore, we are confident that our SOA yields in the absence of $NO_x$ is both useful and representative for the conditions described above. We also measured HOMs at the exit of the flow reactor, but the experimental setup (optimized for SOA formation) did not allow direct quantification*

*of the HOM yields or new detailed insights in alkane HOM formation, except for assessing the role of multi-generational OH oxidation."*

2. Lines 75-80. Is there a specific reason why RH was set to 22%.

   Response: This was simply due to our water bath setup which under these operating conditions could not reach higher RH than 22 %.

3. Line 89-92. Please describe the difference between the measured and modelled OH exposure more quantitatively. In a typical experiment, how big of a difference would this be in calculated precursor consumption values?

   Response: The difference in measured and modelled OH exposure affects the final calculated SOA yields. We have now addressed the issue in more detail and added several uncertainties to account for the differences. This is explained in more detail in Major Comment 1 to Reviewer 2 (further below).

4. Line 113-115. Were wall losses to the PAM and/or tubing corrected for? If not, how can the measured SOA mass concentrations, and thus yields, be trusted if HOM condensation to the wall is a significant issue? Is it possible to lower the oxidant concentrations in the PAM and reduce the number of instruments used at a time in order to reduce HOM yield and sample flow to improve the HOM measurements? If HOMs would be significantly lost regardless of sample flow, how can any particle mass measurements be trusted?

   Response: We appreciate the reviewers for pointing this out. It is challenging to estimate exact losses for low volatile compounds in the PAM, but a good first approximation is the model for estimating the fate of low-volatility organic compounds (LVOCs) by Palm et al. (2016). We have now incorporated such model runs in the revised manuscript. However, with reference to the reviewer's comment, much of the HOMs will be lost to the particles (i.e., the SOA formation itself is a loss for HOM), so any experiment with long residence times will automatically lose the vast majority of HOMs before sampling, making their quantification challenging in most systems. Our setup was designed for aerosol measurements and therefore we repeatedly refer to the study by Wang et al. (2021) for results from a HOM-optimized experiment.

   *Line 115-122: "Correcting for gas phase losses in a system is challenging, however, we applied the model for estimating the fate of low-volatility organic compounds (LVOCs) by Palm et al. (2016). This model account for e.g., wall losses in the PAM and calculates the fraction of LVOCs that are condensed onto particles and forms the SOA we measured with the AMS + SMPS system (described in the section below). In general, over 90% of all LVOC condenses onto particles, with a decreasing trend with increasing $OH_{exp}$ (Fig. B11 as an example output from the model). However, as no seed aerosol is used for these measurements and the model uses condensation sink (CS) as an input parameter, the models does not capture any (wall) losses before nucleation has happened inside the PAM. Therefore, we expect that the real gas phase losses are somewhat greater than the model predicts."*

[Figure]

*Figure B11. Fraction of different fates for LVOCs in the PAM-chamber for one of the n-butylcyclohexane experiments. Model from Palm et al. (2016) is used to produce the figure.*

Particle transmission was already corrected for (although not mentioned) in the original manuscript, and is now described in Section 2.1. as:

Line 111-115*: "To account for both particle and gas phase losses in the PAM, we have applied two corrections. For measuring particle transmission at different sizes, we used dry ammonium sulphate particles ranging from 30 nm to 400 nm in diameter (size selected with a differential mobility analyzer (DMA)). Size dependent particle transmission (Fig. B10) was gained when comparing the particle number concentration before and after the PAM. For particles larger than 80 nm, the transmission was always over 90 %, while it was lower for smaller particle sizes."*

[Figure]

*Figure B10. Particle transmission of our PAM setup as a function of particle size*

5. Lines 148-155 on page 6 and page 7. Please explain why cycloalkanes had the highest SOA yields.

Response: Similar to our results and as already described in the manuscript, previous studies has also found that cyclic compounds have a higher SOA yield compared to acyclic compounds. However, the reviewer is correct that we don't explain enough the reasons behind this behavior. The following section has now been added to the manuscript:

Line 189-195: *The underlying reason for higher SOA yield for cyclic alkanes has already been discussed in previous studies, e.g., in Lim and Ziemann (2009) and (Hunter et al., 2014); during the oxidation step, acyclic compounds have higher risk for fragmentation. Fragmentation of the parent carbon-chain leads to products that are too small and volatile for participating in condensation and SOA formation. During the OH oxidation step, carbon-carbon bond (C-C) scission of linear and branched compounds tends to cause fragmentation, while cyclic compounds can prevent the fragmentation of the parent carbon-chain with ring-openings. Therefore, cyclic moieties can undergo several functionalization steps including oxygen addition before any major fragmentation occurs.*

6. Line 151 on page 7. Authors mentioned that SOA mass concentration were at atmospherically relevant levels ($< 20$ µg/m3) for all precursors. All experiments were under controlled environments and not relevance to atmospheric conditions. The description about SOA mass concentrations is not appropriate.

Response: We do not fully agree with the reviewer on this point, as we did not claim that our experiments were made under atmospheric conditions, but rather that the SOA concentrations were similar to typical values found in the atmosphere, i.e., of atmospheric relevance compared to other studies where the SOA concentrations could range in the hundreds of micrograms. However, we have reformulated the text to say that the SOA concentrations in our experiments were typical of concentrations found in the atmosphere.

Line 185: "*The SOA mass concentrations ranged from 2.5 to 66 µg m$^{-3}$, which is typical mass concentrations found in the atmosphere.*"

Line 320-321: "*producing mass concentrations typically found in the atmosphere ($< 70$ µg m$^{-3}$).*"

7. Lines 174-179. Please clarify the seed types and their role on SOA yields. In the absence of inorganic seed, SOA formation can be retarded but its yield is not affected. The organic seed aerosol can increase SOA yields.

Response: several studies has shown that the SOA yields in the presence of seed can be up to a factor of 2 to 3 higher (Lambe et al., 2015; Ahlberg et al., 2019). This is now mentioned in the manuscript as:

Line 217-220: "*For n-butylcyclohexane, the only reported SOA yield, 38 %, is from Lim and Ziemann (2009) where the experiment was done in presence of both organic seed aerosol and NO$_x$ at over 1800 µg m$^{-3}$ SOA mass concentrations. Their result is therefore not directly comparable to our SOA yield of 17-23 %. However, the presence of seed aerosol can increase the yield up to a factor of 2 to 3 (Lambe et al., 2015; Ahlberg et al., 2019); therefore, our results lie within that range.* "

8. Line 201-206. Authors need better explanation on why this study is not expected to be in the range of ambient SOA studies. Even if they include some biogenic SOA with lower H/C ratios, wouldn't that only lower the bottom of the range without having an impact on the upper limit of the range? What were the oxidation conditions (i.e NOx and oxidant concentration) of the vehicle exhaust studies? Are they comparable with this study?

Response: The shaded area in Fig. 3 represents ambient OOA (according to (Ng et al., 2011; Canagaratna et al., 2015), but as already mentioned in (Ng et al., 2011), hydrocarbon like OA (HOA) does not fall into the "ambient space". As emissions from combustion processes can be classified as HOA, and fossil fuels are mostly consisting of hydrocarbons, we can approximate that our alkane SOA can be seen as HOA. Therefore, it is not surprising that our alkane SOA H/C ratios are higher than the ambient range. While the vehicle exhaust studies were not done under similar conditions as our measurements, we added this range in the Van Krevelen diagram to demonstrate that SOA from this kind of emissions differ from regular ambient SOA.
This is now addressed in the manuscript as:

Line 250-253: "Already Ng et al. (2011) observed that hydrocarbon-like OA (HOA) lies outside the ambient range (higher H/C ratios), which can partly explain why our results shows a similar trend; emissions from fossil fuel combustion, consisting of mainly alkanes, can be classified as HOA."

9. Figure 3. Again, environmental conditions of alkane SOA formation of this study is very different from ambient air. Authors need to explain why alkane SOA of this study is comparable with SOA in ambient air.

Response: This comment is already addressed in the previous comment (comment 8). In addition, this has also been addressed in the response of Comment 1, where we motivate our low-NOx conditions for alkane oxidation.

10. Figure 4. Overall, mechanistic explanations for different HOM yields are weak. For example, among oxygenated VOC, why did different VOCs yield different HOM productions?

Response: Indeed, we do not provide any mechanistic explanations, but this was never the intention of our study. Our measurements were not suitable for mechanistic explanations, so it is an intentional decision to not mention it. To deemphasize the HOM part of this study, we have modified the title and removed the HOM part from it.
The new title of the manuscript is:

*SOA yields from C$_{10}$ alkanes and oxygenates*

We have also modified the manuscript to emphasize that our main goal in this study is to provide SOA yields for the selected VOCs, and any additional HOM measurements are a bonus. Our HOM measurements at a minimum show that all of our 7 VOCs can produce detectable amounts of HOM. However, Figure 4 was added to show that there could be a correlation between HOM and SOA for alkanes. To deemphasize the HOM vs SOA correlation from our study, we have moved the figure to the Appendix.
Although the experimental setup between our study and Wang et al. (2021) are not directly comparable (as already stated in the manuscript), we can see that the four VOCs that showed measurable HOM yields from Wang et al. (2021), are the same VOCs that resulted in the four highest SOA yields in our study (excluding 2-decanone which was not studied in Wang et al. (2021)). This comparison is far from perfect, but is the first comparison between alkane HOM and SOA. Our intention with this figure is also to encourage any upcoming measurements; with a different experimental setup, a better comparison is possible.

The concern for too little discussion and analysis of the HOM data is also discussed in the response of Major Comment 2 to Reviewer 2.

11. Figure 5. The reviewer cannot understand the role of Figure 5 to explain the formation of HOMs. Naturally, higher concentrations of precursor with a given concentration of oxidants will produce less oxidized products and vice versa.

Response: Clearly, we had not described the figure well enough, as the reviewer seems to have misunderstood what it is supposed to show. We have now added text to make it clearer. The idea here is to show that the average H/C ratio decreases with increasing VOC concentrations, which indicates that the HOMs are formed in increasing amounts from multi-generation OH oxidation. We do not claim that this is unexpected, but rather simply wanted to confirm that even our non-optimized HOM observations can capture this trend. The findings is similar to the studies done with aromatic compounds in Garmash et al. (2020).
Note: as Fig 4 from the original manuscript is moved to the Appendix in the revised manuscript, the figure numbering is different in the new version.

Line 291-293: "Although our setup was not optimized for HOM detection, we were able to capture the behaviour of decreasing average H/C ratio with increasing VOC concentration. These findings are similar to studies done with aromatic compounds in Garmash et al. (2020)."

12. Line 227-231. HOMs are discussed in the introduction in relation to autoxidation (line 46). Typically, the RO radical that initiates the alkane autoxidation process is formed in the reaction with NO. Discuss the significance of the RO2 + RO2 reaction in no NOx conditions.

Response: Similar to RO2 + NO reactions, the main products of RO2 + RO2 reactions are typically also RO radicals, and thus RO-initiated autoxidation can occur in a very similar way in both systems. Differences in exact yields of RO radicals will vary between the two reaction pathways, but will also vary for the same pathways depending on the structure of R. The yields of organic nitrates from RO2 + NO can vary strongly, while yields of accretion or alcohol/ketone products can similarly vary for RO2 + RO2. We have added a note on this in the revised manuscript.

Line 301-306: "As discussed in previous sections regarding the SOA formation, our measurements were done in the absence of $NO_x$, which can change, in addition to SOA yields, the gas phase products and reactions pathways. However, similar to $RO_2 + NO$ reactions, the main products of $RO_2 + RO_2$ reactions are typically also RO radicals, and thus RO-initiated autoxidation can occur in a very similar way in both systems. Differences in exact yields of RO radicals will vary between the two reaction pathways, but will also vary for the same pathways depending on the structure of R. The yields of organic nitrates from $RO_2 + NO$ can vary strongly, while yields of accretion or alcohol/ketone products can similarly vary for $RO_2 + RO_2$."

13. Line 231-235. It would be beneficial to the reader to provide a mechanism where oxidation is initiated and proceeds through the major reactions that are expected to occur under these conditions which ultimately lead to HOM (>6 Oxygen atoms) identified in Figs. A2-A8.

Response: As explained in the response of Comment 10, our intention is not to provide any mechanistic explanations as our experimental setup was not optimized for that. See the response of Comment 10 for details.

14. What is the impact of heterogeneous chemistry on SOA yields?

Response: Heterogeneous reactions should not be significant at our OH exposure levels (equivalent of < 2 days of aging) (Cappa and Wilson, 2012; Chen et al., 2013). Therefore, it is not considered or accounted for in our study.
A note of this is now added to the manuscript:

Line 123-125: *"Previous studies (Cappa and Wilson, 2012; Chen et al., 2013) have shown that heterogeneous reactions should not be significant at less than two days of ageing. Therefore, we do not account for any heterogeneous chemistry in our SOA yield calculations as all our measurements were done under one day of atmospheric ageing."*

15. Appendix A. There is a dilution flow (10 LPM). What is the impact of dilution of air stream on SOA production? Some SOA products can be off-gassing from the aerosol. Please clarify this issue.

Response: A dilution flow can affect the SOA, but without measuring the direct effect, we can not asses the magnitude of it. However, we added an uncertainty of $\pm$ 30 % to the SOA measurements to account for different uncertainties in the particle mass measurements by the SMPS, and given the very short time available between the dilution and sampling, we expect that off-gassing will be a minor contributor to this uncertainty.
Details of added uncertainties to the SOA yield calculations can be found in the response of Comment 1 from reviewer 2.

16. Figures A2-A8. What is the signal to noise (S/N) ratio? In general, the signal should be 3 times higher than the noise. Were elemental compositions marked on Figures statistically significant.

Response: As the resolution of the instrument is high (~ 8000), all the peaks identified and marked in the figures are fitted in high resolution and with confidence.

**Reviewer: 2**

**Review Report Graeffe et al. 2025: SOA yields from C10 alkanes and oxygenates and their relation to highly oxygenated organic molecules (HOM)**

Graeffe et al. present a study about the Secondary Organic Aerosol yield from a number of C10 compounds from photooxidation in the presence of ozone and under very low NOx concentrations in an Oxidative Flow Reactor (OFR). The list includes n-alkanes, branched, mono and bicyclic structures as well as the oxygenated compounds (aldehyde, ketone, alcohol). They find the expected trends for the SOA yields and try linking them with the observed Highly oxygenated Organic Molecules (HOM).

Such systematic, fundamental work furthers our understanding of SOA formation mechanisms in the atmosphere and is thus of interest for the audience of this journal. However, I found one important weakness in their methodology (the estimation of the VOC concentration, see Major Comment #1). Before publication, the impact of this issue needs to be discussed in detail to fully evaluate the findings of this study.

**Major comments**

1) Knowing the amount of reacted VOCs is crucial for the calculation of aerosol yields. Apparently due to some instrumental misfortune, the VOC concentrations (ingoing and outgoing) could not be measured for most of the experiments. The authors rely on the calculated concentrations from the injection method (flow of syringe pump) for the initial VOC concentration in the OFR. They then calculate the reacted amount from the OH exposure and known reaction constants. This method in itself is a valid approach if no measurements are available. But there are several sources of uncertainty for the calculated reacted VOC concentration which is the crucial parameter for the yield calculations.

a. Uncertainty of the initial VOC concentration. Do the authors have any indication of the accuracy of this estimation? E.g. how do the calculated and measured values of the initial VOC concertation compare for a case where they do have PTR data? (e.g. the OH exposure experiment with nonanal)

Response: Unfortunately, the PTR was not calibrated towards nonanal prior the experiments, so we can not get any quantitative results from it. As the main purpose, initially, was to use the PTR for OH exposure calibration, where only the relative change of the measured signal is important, we do not have any PTR data to support our VOC concentration estimation.
However, we did quite extensive calibration of the syringe pump setup, as we tested several syringes to find one that would both last long enough (i.e., not be empty of VOC in the middle of an experiment) as well as inject the VOC in a suitable injection rate without any other practical issues. Therefore, we know exactly how much of the VOC (in liquid form) was injected in the VOC carrier flow. By knowing the volume of the injected VOC, we were able to calculate the concentration in the flow. In addition, as VOCs per definition are volatile, we do not see any reason why most of the injected VOC would not be transported to the OFR, instead of sticking on the tubing. The transmission efficiency of the VOC should be near unity, but we will add an uncertainty of ± 5 % to the error calculations to account for errors.
More generally, we have routinely used this method for injecting monoterpenes and other alkenes in our lab (Meder et al., 2023; Zhao et al., 2024) and the calculated concentrations are always in good agreement with PTR observations.

b. OH exposure. The authors use the OH estimator model from Li et al. (2016). They compared the calculated value with a test measurement for nonanal (a compound not used in the actual yield experiments) and report a more than two times higher OH exposure value from the measurements than from the model. Doing a simple calculation using a k(OH) of 1e-11 cm3 molec-1 s-1, I calculate that the higher OH exposure leads to a 1.8 times higher amount of consumed VOC (see table below). The yield would be affected by the same factor.

*Table 1: Calculation of consumed VOC with measured and modelled OH exposure*

| | | |
|---|---|---|
| OHexp(meas) | 6.20E+10 | molec. /cm3/s |
| OHexp(model) | 2.90E+10 | |
| | | |
| VOC initial | 35 | ppb |
| k | 1.00E-11 | cm3/molec./s |
| | | |
| VOC | consumed | |
| meas | 16.2 | ppb |
| calc | 8.8 | ppb |
| | | |
| meas/calc | 1.835474 | |

Since the authors use the model for all experiments, potentially all VOCconsumed values are too low. Then the calculated yield would be too high. But this is just my first "top-of the- head" thought and the authors need to go into the details.

Response: We want to thank both reviewers for pointing this out as this is probably the largest source of uncertainty in the SOA yield calculations, and there was a need to address this in more detail than in the original manuscript.
Firstly, we found an error in our calculation of $OH_{exp}$ from the nonanal measurements and model, which changed the measured $OH_{exp}$ to $5.5 \times 10^{10}$ molecules $cm^{-3}$ s (from previous $6.2 \times 10^{10}$) and modelled $OH_{exp}$ to $3.5 \times 10^{10}$ molecules $cm^{-3}$ s (from previous $2.9 \times 10^{10}$).
Secondly, the reviewers' calculations of the difference between the measured and modelled $OH_{exp}$ get too large due to too low rate constant; the reviewer used k(OH) of $1 \times 10^{-11}$ instead of $3.6 \times 10^{-11}$ which is found in the literature (Bowman et al., 2003). However, the reviewers' point stands, the modelled $OH_{exp}$ is 37 % lower than $OH_{exp}$ derived from the nonanal calibration. Therefore, the difference in the consumed VOC ($\Delta VOC$) is still significant: $\Delta VOC(measured)/\Delta VOC(modelled) = 1.20$. This will further have an impact on the calculated SOA yields.
However, we still decide to use the modelled $OH_{exp}$ as it can capture the changes that comes from different VOC concentrations etc.
The other uncertainties described in these sub sections (i.e., initial VOC concentration, residence time and rate constants) will affect the modelled $OH_{exp}$, but we will add an extra uncertainty due to the discrepancy between the modelled and measured $OH_{exp}$.
Therefore, we will add an uncertainty of ±20 % to $OH_{exp}$ when doing error calculations for SOA yields.

c. Residence time. It was not said explicitly, but I assume that the stated residence time was derived assuming plugged flow. PAM characterisation papers showed that due to the inlet geometry, the flow inside PAM is not represented by the "plugged flow" simplification. Fig 3 in (Lambe et al., 2011) shows an example of how much plugged flow and the actual residence time distribution can differ. The residence time is not used directly in the calculation, but it is folded into the calculated OH exposure where one single value is used for the residence time. Do the authors have any measurements for the residence time distribution in their PAM with their specific flow setup? What would the plugged flow residence time represent? Do most VOC molecules experience a shorter residence time?

Is the plugged flow a representation of the average residence time? How would this affect the calculated vs measured OH exposure and thus ΔVOC?

Response: Yes, the residence time was derived assuming plugged flow. Unfortunately, no measurements for the residence time distribution (RTD) were done with the system.
Without any measurements, we can only do qualitative assumptions. However, it is clear that the plug flow will not represent the real RTD in the PAM. We can assume that the shape of the RTD would be similar to the one in Lambe et al. (2011) (Fig 3 as the reviewer mentioned). This would mean that the majority of the VOC molecules would have shorter residence time, but a significant part of the VOC would also have much longer residence time than the assumed plug flow time.

We tested how changing the residence time would affect the final SOA yields by doubling and halving the used residence time (i.e., 80 s). In this test, by changing the residence time, it would first affect the $OH_{exp}$, then affect ΔVOC and finally the SOA yield. It seems that the effect on the SOA yield when of changing the residence time is quite limited in our PAM conditions. Halving and doubling the residence time still gave the same SOA yield within ± 5% of the originally calculated. Therefore, we can assume that the plug flow is a good proxy for the residence time in our experiments. However, we added an uncertainty of ± 10 % to the residence time when doing error calculations for the SOA yields.

d.   Reaction constants. What values were used for the reaction coefficients? I did not see any values or reference. Like any other parameter, reaction coefficients will have an uncertainty which will directly contribute to the uncertainty of the calculated ΔVOC concentration. From the typical values found in literature for these type of compounds (e.g. (Shaw et al., 2020)), I would guess a 10-20% uncertainty in the reaction coefficient values could be reasonable. How much does that add to the overall uncertainty for ΔVOC and thus the uncertainty of the aerosol yields?

Response: The rate constant, with references, are now added to Table 1. Regarding the uncertainty of the constants, we have included a ± 15 % uncertainty of the constants to the ΔVOC calculations.

With all this in mind, the authors should provide a thorough discussion of the overall uncertainty of their estimated VOC concentration values and how that will affect the calculated SOA yields. E.g., if the reported values are a lower or higher estimate and how much additional uncertain stems from the VOC estimation. Such uncertainty needs to be taken into account in the discussion when comparing the yield values from this study with literature values.

Response: with the answers in sub comments 1a-1d, we have now added several uncertainties to our calculations that are needed for the SOA yields. In addition to the above-discussed points, we added also an uncertainty to the particle mass measurements by the SMPS.

        Here is a summary of the uncertainties that are now used:

- Initial VOC concentration: ± 5 %
- OH exposure: ± 20 %

- Residence time: ± 10 %
- Reaction rate constants: ± 15 %
- SOA measurements: ± 30 %

We added a section in the appendix regarding the error calculations:

Line 377-413: **"Appendix B: Corrections and uncertainties for calculating SOA yields**

As described in Section. 3.1.1., the SOA yields (Y) are calculated as the ratio of formed organic aerosol concentration ($C_{OA}$) to reacted precursor concentration ($\Delta VOC$) by:

$$Y = C_{OA}/\Delta VOC \tag{1}$$

, where $\Delta VOC$ is:

$$\Delta VOC = [VOC] \times \left(1 - e^{-k \times OH_{exp}}\right) \tag{2}$$

, where k is the second-order rate constant of the precursor with OH and [VOC] is the injected VOC.

The following uncertainties has been applied in the error calculations:

- initial VOC concentration: ± 5 %
- PAM residence time: ± 10 %
- reaction rate constant: ± 15 %
- $OH_{exp}$: ± 20 %
- SOA measurements: ± 30 %

Initial VOC concentration is crucial for calculating SOA yields, and it was calculated from the injected volume by the syringe pump. Unfortunately, the PTR-ToF was neither functioning throughout the measurements or successfully calibrated, so we cannot compare any measured VOC concentrations with the calculated one. However, the syringe pump was calibrated prior the measurements, and the injected volume can be trusted. Despite that, the add an uncertainty of ± 5 % to the initial VOC concentration.

The PAM residence time is needed in the $OH_{exp}$ model by Li et al. (2016), where we assumed ideal plug flow when calculating the residence time. However, in e.g., Lambe et al. (2011) the actual measured residence time distribution (RTD) can differ significantly from the ideal plug flow. As we did not perform any measurement of the RTD for our setup, we assume that our RTD is similar to the one in Lambe et al. (2011). By testing how the SOA yield changes with different modelled $OH_{exp}$ (by increasing and decreasing the residence time), we noticed there is only small changes in SOA yield (< 5 %) even if the residence time would be ± 20 %. Therefore, we conclude that our calculated residence time by assuming ideal plug flow is a good proxy for our setup. However, we still add an extra uncertainty of ± 10 % for the final error calculations.

Reaction rate coefficients are needed in both the modelled $OH_{exp}$ and the $\Delta VOC$ calculations. However, there is always some uncertainty in these measured values (Atkinson, 2003). Therefore, we have added an extra uncertainty of ± 15 % to account for this.

Additional uncertainty comes with modelled $OH_{exp}$ as we do not have calibration measurements over the whole range of conditions used in this study. While the measured $OH_{exp}$ from the nonanal calibration is higher than the modelled $OH_{exp}$ for the same conditions, we decided to use the modelled $OH_{exp}$ for all our calculations as it can capture the changes in different condition for the different SOA precursors. All of the abovementioned parameters (initial VOC concentration, PAM residence time

*and rate constants) are already included in the OH$_{exp}$ model. However, we will add an additional uncertainty of ± 20 % to account for any other uncertainties arising from the modelled OH$_{exp}$.*

*To account for the AMS and SMPS derived SOA mass, we use an uncertainty of ± 30 %. This includes the uncertainty of the both particle phase instruments, as well as the uncertainty from the LVOC fate model (described in Section 2.1)"*

Due to these changes, Table 1 has been modified in the revised manuscript as:

*"Table 1. Summary of all the VOCs (including name, molecular formula, structure and rate constant with OH), amount of injected and reacted VOC, formed SOA mass and their SOA yields.*

| Compound | Molecular formula | Structure | Rate constant k (cm$^3$ molecule$^{-1}$ s$^{-1}$) | Injected VOC (ppb) | Reacted VOC (ppb) | Formed SOA-mass (µg m$^{-3}$) | SOA yield (%) |
|---|---|---|---|---|---|---|---|
| **Alkane** | | | | | | | |
| n-decane | C$_{10}$H$_{22}$ | | 1.1×10$^{-11}$ [a] | 39-98 | 19-29 | 4.8-9.3 | 4.4-5.5 |
| 2,7-dimethyloctane | C$_{10}$H$_{22}$ | | 1.1×10$^{-11}$ [b] | 29-120 | 16-32 | 2.5-8.4 | 2.8-5.2 |
| n-butylcyclohexane | C$_{10}$H$_{20}$ | | 1.47×10$^{-11}$ [c] | 14-110 | 10-33 | 11-39 | 17-23 |
| cis-decalin | C$_{10}$H$_{18}$ | | 2.01×10$^{-11}$ [d] | 16-79 | 12-30 | 22-66 | 32-39 |
| **Oxygenate** | | | | | | | |
| decanal | C$_{10}$H$_{20}$O | | 3.25×10$^{-11}$ [e] | 13-52 | 11-27 | 4.1-26 | 6.0-15 |
| 2-decanone | C$_{10}$H$_{20}$O | | 1.32×10$^{-11}$ [f] | 43-120 | 21-33 | 4.4-20 | 3.4-10 |
| 1-decanol | C$_{10}$H$_{22}$O | | 1.5×10$^{-11}$ [g] | 120-360 | 33-41 | 7.8-24 | 3.6-9.1 |

(a) Atkinson (2003)

(b) estimate (Parchem, 2025)

(c) Atkinson and Arey (2003)

(d) Atkinson et al. (1983)

(e) estimate, average from Wang et al. (2021) and Pubchem (2025a)

(f) Wallington and Kurylo (1987)

(g) estimate (Wang et al., 2021; Pubchem, 2025b)"

In addition, Figure 2 and its error calculation is modified according to the changes described above:

[Figure]

*Figure 1. SOA yield as function of (a) injected VOC, (b) reacted VOC and (c) formed SOA. The shaded area represents the error as described in Appendix B.*

Furthermore, Fig. 4 (Fig. in the original manuscript) is also modified:

[Figure]

*Figure 4. Ratios indicating the importance of multi-generation OH oxidation. The ratios on the y-axis are presented as $C_{10}H_nO_y/C_{10}H_{n+2}O_y$, with y=4-9 and n=14 for cis-decalin, n=16 for butyl-cyclohexane, 2-decanone and decanal, and n=18 for n-decane, 2,7-dimethyloctane and 1-decanol. The inner color (white to black) corresponds to the reacted VOC (%) of the data point. For visualization, the colorscale range is set from 20 % to 90 %, which includes all precursors except 1-decanol that ranges from 11 % to 28 %.*

2) The HOM part is too superficial in my opinion. I understand that the experiment design was not favourable for quantitative HOM measurements. But only two C10 HOM groups seem to be investigated in the text. In the appendix figures A2-A8, 6-8 ions (all C10) are identified. These do not seem to be the most prominent (strongest) signals in that m/z range. Why were these chosen? Are these identified HOM representative for the overall HOM population? Does the molar HOM yield in Fig 4 include all potential HOM in the system or also only a specific subset (e.g., only C10 compounds)? (See also specific comment 24)

Response: We decided to investigate only C10 compounds in this study as the gas phase data had limitations, and these limitations are also the reason for us only being able to discuss our own HOM results on a superficial level. Instead, we utilize the findings by Wang et al. (2021) for most comparisons of HOM formation and yields. In their study, they studied all HOM species, but as those were primarily first generation oxidation products, the HOMs were predominantly composed of molecules that retained the same amount of C-atoms as the precursor alkanes.

For our study, we assume that most of the HOMs formed inside the PAM are lost to condensation (SOA formation) or walls. We do not expect these compounds to survive all the way to the NO3-CIMS. However, we believe that the HOMs we detect are formed in the very end of the PAM chamber or in the tubing leading to the NO3-CIMS. Due to this, we do not do any quantitative analysis. Furthermore, we did observe C7-C9 compounds, but decided to concentrate on the C10 compounds, as we do not expect us to have the required data to anyway do more detailed mechanistic explanations (e.g., fragmentation, which should be more likely for acyclic compounds). Another reason for not going into too much detail about gas phase data, is the experimental setup. As the conditions in the PAM chamber does not represent atmospheric conditions, we do not want to draw too much conclusions of the detailed gas phase reactions happening in the PAM.
Investigating the role of first and second generation OH oxidation was done by only comparing C10 compounds. This approach should be sufficient to address this issue and show the decreasing trend in the $H_n:H_{n+2}$ ratio.

In my opinion, the authors could strengthen the scientific impact of this manuscript by expanding the discussion of the HOM compounds.

Response: This is true, but we want to emphasize that our aim in this manuscript was never to go into detail in HOM processes, formation or mechanistic explanations. Therefore, we have now modified the title of the manuscript and removed the HOM part from it to directly show that our emphasis is on the SOA yields.
The new title of the manuscript is:

*SOA yields from $C_{10}$ alkanes and oxygenates*

In addition, as our setup was not optimized for HOM detection, we recognize the limits of the HOM data and do not want to be too speculative and draw any conclusion that we can't support. Therefore, we have now modified the manuscript to clearly state that our goals lie in the SOA yield measurements and the HOM measurements are just a bonus that can strengthen our claim that HOMs can be important in alkane SOA formation (as we were able to detect HOMs for all precursors). This is also discussed in more detail in the response of Comment 10 from Reviewer 1 and the response to the comment above this one.

**Specific comments**

1) Line 48: what is the "bimolecular reaction rate" referring to? O2 addition to the alkyl radical? Or the initial OH+VOC reaction?

   Response: Our intention was to explain that the oxygen content in oxidation products generally increased when more bimolecular reactions took place. This is motivated by Fig 4a in Wang et al. (2021), where increased NO concentration leads to higher molar yield (of products with 4,5 or 6 oxygens). Due to their experimental setup, most RO2 radicals would just exit the flowtube without reacting further (due to short residence time) when no NO was present. By adding NO, bimolecular reactions accelerate the RO2 radical conversion via RO2 + NO → RO + NO2. And as described in the manuscript (right after the sentence in question), RO radicals are important in the adding of oxygen to the compound.
   We have now rephrased the manuscript as:

   Line 46-48: *"Wang et al. (2021) did not only measure HOM yields, but showed that the oxygen content in oxidation products generally increased* when more bimolecular reactions took place, *even though not always reaching six or more O-atoms."*

2) Lines 72-74: The description of the oxidant formation is not strictly precises. OH is not produced from water vapour, but from photolysis of O3 and then consecutive reaction of O(1D) with H2O. HO2 radicals are mostly formed from photolysis but from the reaction of OH with O3 (or VOCs). HO2 is listed as an "oxidant" in the same way as OH is. But to my knowledge. HO2 will not initiate oxidation reactions with the investigated VOCs but rather participate in the reaction mechanism, e.g. terminating RO2 radical chemistry. This sentence needs rephrasing and clarification.

   Response: We have now rephrased the section regarding oxidant formation in the PAM as:

   Line 83-89: *"Ozone ($O_3$) is primarily produced from the photolysis of molecular oxygen: $O_2$ + hv (185 nm) → 2 $O(^3P)$, followed by $O_2$ + $O(^3P)$ → $O_3$. Hydroxyl radicals (OH) are primarily produced from $O_3$ + hv (254 nm) → $O_2$ + $O(^1D)$ followed by $O(^1D)$ + $H_2O$ → 2 OH.*

3) Line 76 & Fig A1: To my knowledge, most PAM systems use a Nafion humidifier and not a water bath for humidification.

   Response: We used an older model of the PAM and did not have a Nafion humidifier available, so we used an external water bath to increase the humidity. However, due to limitations of the water bath setup, we could only increase the RH to 22 %.

4) Line 76: 22% RH is a rather low value. Often 40% RH are used in OFR or chamber studies as a representation of average atmospheric conditions. Furthermore, the recommendation to obtain atmospherically comparable VOC and RO2 chemistry for OFR185 mode is to use high H2O, low UV, and low OHRext (see conclusions of (Peng et al., 2019)). Was there a reason for choosing 22% or did this stem from the limitation of the setup (i.e., the efficiency of the humidifier at the high flow needed for the instrumentation)?

Response: As the reviewer correctly guessed, this was mainly a limitation of our setup and the high flows needed. RH 22 % was the maximum value due to the setup limitations as explained in the response of the previous comment.
However, $OH_{ext}$ was mainly below 40 s$^{-1}$ (except for 1-decanol) which is moderate (if 10 s$^{-1}$ is low and 100 s$^{-1}$ high, (Peng et al., 2015)) and the UV lamps are also on a moderate level (100 V out of 190 V). Although the RH is lower than ideally, we estimated that the conditions in the PAM are reasonable for obtaining atmospherically relevant chemistry.

5) Line 79: The UV lamps were set to 100V? To my knowledge PAM UV lights use a control voltage of 0-10 V. Was a different system used? Or is this a different voltage?

Response: We assume the 0-10 V voltage control is true only for the newer PAM chambers. However, our (older) PAM system is the one used in for example Karjalainen et al. (2016), where the UV lamps are controlled by external voltage controllers from 0 to 190 V.

6) Line 77: I assume the residence time value is calculated assuming plugged flow. This needs to be clarified.

Response: Yes, we assumed plugged flow. As explained in the response of Major Comment 1c, this is now addressed in the uncertainty calculations.

7) From the description, I assume PAM was operated in "OFR185" mode, i.e., using the 185 nm UV lamps inside PAM to generate O3 and OH at the same time? Using this label may help clarify the mode of operation and facilitate comparison with other studies.

Response: Indeed, we operated the PAM in OFR185 mode, and this is now explicitly mentioned in the PAM section.

Line 86-87: "The PAM was operated in OFR185 mode, meaning that UV lamps inside the PAM emitted at two wavelengths, 185 nm and 254 nm."

8) Line 111 and Figure A1: I understand that additional dilution after PAM was necessary to achieve enough sample flow. The sampling flows in Fig A1 add up to 14 lpm. Why was the dilution flow set to 10 lpm creating an overflow of 6 lpm?

Response: Originally, we had plans to have more instruments connected to the setup. For example, we had a NOx monitor connected (to verify that we were in extremely low-NOx conditions), which needed ~3 LPM, but similarly as the PTR-TOF, it was not operational for the whole experiment and was not mentioned/included in the manuscript.
Therefore, the original design of more dilution stayed as we wanted to keep the dilution flow constant throughout the whole experiment.

9) Line 89f: I do not understand how the authors come to the conclusion that the measurements do not include the external reactivity. The presence of nonanal in the measurements is the "external OH reactivity". The derived OH exposure values will be the values in the presence of the set amount of nonanal.

I assume that the authors used the "OFR exposure estimator" based on Li et al. (https://sites.google.com/site/pamwiki/estimation-equations). Then the reason that the model estimates a lower value is that it assumes that the set external reactivity is present for the whole residence time. This is equivalent with assuming that the 35 ppb of nonanal are never consumed. Or that the reaction products of nonanal continue to react with OH at the same rate as the precursor and that they continue to do so for the whole residence time. Neither of these are of course correct – the truth is somewhat in the middle (precursor gets consumed, reaction products react). However, the discrepancy between the model and measurements does not stem from "not include the external reactivity".

Response: We appreciate the reviewer's comment. The reviewer is in indeed correct and this is a mistake from our side. We have now deleted that part.

10) Line 96: If the authors provide the information about the AMS data acquisition interval, they should also mention if they were only alternating between open and closed or if they also ran pToF. Typical settings are 20sec for open/closed. Then only 2/3 of the 1 min would actually contain MS data.

Response: No pToF data were collected as we relied on the SMPS data for sizing data. In addition, we did not trust our pToF measurements as we have had problems with it previously. This is now described in the manuscript:

Line 129*: "The AMS was alternating between open and closed mode with 1 min acquisition"*

11) Line 97: what is meant with an "overflow of 1L/min"

Response: The AMS tales only 0.1 LPM inside the instrument, but we increased the sample flow from the PAM to the AMS to 1 LPM to minimize particle losses in the tubing. 0.9 LPM was pulled using an external vacuum pump until the aerodynamic lens in the AMS. This is now rephrased in the manuscript as:

Line 130-131*: "Sample flow from the PAM to the AMS was set to 1 L min$^{-1}$ with an external vacuum pump, with only 0.1 L min$^{-1}$ being sampled into the AMS."*

12) Line 102: what is meant by "area concentration"?

Response: By getting the particle number size distribution from the SMPS, and assuming spherical particles, we can calculate the total surface area of the whole aerosol population. This is what we call area concentration in the manuscript. However, as this parameter is not directly necessary for this work, it is not mentioned anymore in the revised manuscript.

13) Line 104: How much uncertainty is introduced by using this density estimation? If AMS did acquire pToF data, do the estimated densities match the densities that can be derived from comparing aerodynamic and electromobility diameters (at least the trends between experiments)?

Response: No particle sizing data was obtained from the AMS measurements. The density estimation

equation from Kuwata et al. (2012) introduces some uncertainty, but as our measured H:C and O:C ratios are roughly within the range used in Kuwata et al. (2012), we expect our estimated densities to be within 12 % (as calculated in Kuwata et al. (2012)). Larger uncertainty comes from the VOC concentrations and the parameters included in that (as described in previous comments and responses).

14) Line 102: The upper limit of the SMPS was 500nm (electromobility). How did the volume size distribution spectra look? Was there considerable mass/volume at the highest size and could this be an indication that some particles >500nm were omitted in the SMPS measurements? Few particles at that size can already contribute a large portion of the aerosol mass.

Response: Number concentration was highest at sub 100 nm particles, with only a few particles over 300 nm. Therefore, we are sure that we were able to measure all formed SOA. Below are an example of the number and mass size distribution from one of the decalin experiments, which demonstrates that the formed particles are nicely below the upper detection limit of the SMPS.

[Figure]

*Figure 2* Particle number size distribution from one of the cis-decalin experiments.

[Figure]

*Figure 3* Particle mass size distribution from one of the cis-decalin experiments

15) There is no mention of any O3 scrubber being used. Was this indeed the case?

Response: No ozone scrubber was used in the setup.

16) Line 155ff: I do not agree with the interpretation of SOA volatility from the yield data in this way. To me the point is that decaline produces a larger fraction of low volatility material already at low VOC conc. The acyclic ones produce less of those compounds. Hence, less SOA is formed. But the volatility of the condensing products cannot be derived from this. E.g., let's assume that the system allows particle phase partitioning of compounds with C*<1e-4 ug/m3. VOC 1 has 10% of its oxidation products with C*=1e-4 ug/m3 and nothing below that. VOC 2 has 1% products at C*=1e-6 ug/m3. VOC 1 will form more SOA than VOC 2. At low precursor concentrations, VOC 2 may not form any SOA as it does not create high enough concentrations to initiate nucleation (wall losses may also play a role). At higher precursor concentrations, SOA is formed in both cases. But the volatility of the SOA would be higher for VOC 1 then for VOC 2 – so the opposite of what would be expected from the SOA yield.

A real example for a precursor with lower aerosol yield and also lower SOA volatility is the comparison of SOA from farnesene and a-pinene in (Ylisirniö et al., 2019).

My point here, aerosol yields cannot be interpreted directly into SOA volatility in this way without further composition information or information about the volatility distribution of the gas and particle phase products.

Response: We thank the reviewer for the insight and nice example. Indeed, we cannot say anything about the volatility, except that cycloalkanes are more prone to produce enough low volatility products to form more SOA. We have modified this part in the revised manuscript as:

Line 196-200: *"These results indicate that the oxidation of cycloalkanes produces more efficiently low volatility vapours that can condense and form SOA, compared to acyclic compounds. Furthermore, without seed aerosol, the oxidation products need to have low enough volatility to initiate nucleation already at low precursor concentration. In contrast, the steep increases in SOA yields for many other precursors in Fig. 2b suggests that partitioning into the aerosol phase is strongly enhanced as the amount of products (and the SOA mass concentrations) increase."*

17) Section 3.1.2: was the basic parametrisation (Aiken et al., 2007) or the "improved" parametrisation (Canagaratna et al., 2015) used to derive O:H and H:C values from AMS? This should be stated in the methods section.

Response: Parametrization from Canagaratna et al. (2015) was used. This is now mentioned in the text.

Line 232-233: *"The oxygen to carbon (O/C) and hydrogen to carbon (H/C) ratios are calculated according to the improved ambient method described in Canagaratna et al. (2015), and plotted in a Van Krevelen diagram…"*

18) Line 191ff: The trends of O:C with overall yield seem to suggest that higher yields are linked to higher O:C values. But the decanal points are all >0.5 while decaline and butylcyclohexane shows O:C values that go much lower. But all decaline and butylcyclohexane points have a higher yield than even the highest decanal points. With this in mind, can the authors really make this claim about the trends?

Response: The reviewer is correct, just saying that high O/C ratio indicates higher SOA yield is wrong. We have now also calculated and plotted the average oxidation state if carbon ($OS_C$) as the reviewer suggested in the comment below. These plots reveal that pure alkanes have quite stable SOA yield regardless of the $OS_C$ compared to the oxygenated compounds, that shows a higher SOA yield at lower $OS_C$. At same oxidation state, the high-yield compounds (cyclic and decanal) shows higher SOA yields, separating these three compounds again from the low-yield compounds.
Including these plots (together with the already existing Van Krevelen diagram), we feel as they now provide nice additional information of the SOA and can be compared to both already done and future measurements.

The whole section has undergone several changes, and is now:

Line 223-255:

"**3.1.2 *Van Krevelen diagram and carbon oxidation state***

[Figure]

Figure 4. (a) Van Krevelen diagram showing the H/C plotted against O/C. The inner color (white to black) corresponds to the SOA mass concentration of the data point. The red dashed lines correspond to average carbon oxidation states ($OS_c$) and the grey dashed lines are different slopes to guide the reader. The yellow shaded are represents the ambient H/C and O/C range of OOA according to Ng et al. (2011) and improved by Canagaratna et al. (2015). The area within the black dashed line represents data from recent vehicle emissions studies (Zhang et al., 2021; Ghadimi et al., 2023; Hartikainen et al., 2023). Average carbon oxidation state plotted against (b) SOA mass concentration and (c) SOA yield. The inner color (white to black) corresponds to the photochemical age (in hours) of the data point.

The oxygen to carbon (O/C) and hydrogen to carbon (H/C) ratios are calculated according to the improved ambient method described in Canagaratna et al. (2015), and plotted in a Van Krevelen diagram in Fig. 3a. The average carbon oxidation state ($OS_C$) is a parameter that is proposed to describe better the degree of oxidation than pure O/C ratios (Kroll et al., 2011), and can be approximated as $OS_C \approx 2 \times (O/C) - (H/C)$. $OS_C$ is plotted against formed SOA mass concentration in Fig. 3b and SOA yield in Fig. 3c.

In general, for each individual compound, we clearly see that with increasing SOA mass concentration, the O/C ratio and $OS_C$ decreases and the H/C ratio increases (Fig. 3a and b). This is in good agreement with previous studies (Shilling et al., 2009; Kuwata et al., 2012; Day et al., 2022). At low loadings, only the least volatile species, which are generally the most oxidized, are able to condense and form SOA, but as the particle mass increases, more volatile (, less oxidized and longer photochemical aged) compounds can condense onto the particles, leading to decreased O/C and $OS_C$ and increased H/C.

The H/C ratios of the SOA follows roughly the same order as the H/C ratios of the precursors (ranging from 1.8 to 2.2). Furthermore, our elemental ratios and slopes for cis-decalin and n-decane SOA are similar to those in Li et al. (2019).

Figure 3c reveals an interesting feature of the alkane compounds (linear, branched and cyclic): the SOA yield is relatively stable even if the $OS_C$ increases. This is different to the oxygenated compounds, that exhibits a stronger trend with higher SOA yields at lower $OS_C$. Especially cyclic compounds are able to maintain high SOA yield regardless of the oxidation state, indicating again that these compounds are efficient on producing low-volatile oxidation products for SOA formation.

Only the high-yield compounds go into the ambient H/C and O/C range (yellow area in Fig. 3a, (Canagaratna et al., 2015)), while the other compounds have higher H/C ratio than the ambient range. The precursors in our study have quite high H/C ratio, which could explain why they are above the ambient H/C range. Already Ng et al. (2011) observed that hydrocarbon-like OA (HOA) lies outside the ambient range (higher H/C ratios), which can partly explain why our results shows a similar trend; emissions from fossil fuel combustion, consisting of mainly alkanes, can be classified as HOA. Furthermore, all data, except cis-decalin, falls into the area that represents recent vehicle exhaust studies (Zhang et al., 2021; Ghadimi et al., 2023; Hartikainen et al., 2023). As alkanes are

*mostly anthropogenic emissions, these studies represent our experiment better than most typical ambient SOA studies, which can include biogenic SOA from precursors with typically much lower H/C ratios."*

19) Section 3.1.2: Both H:C and O:C values vary for all precursors. Did the authors look into using OSc (average oxidation state of carbon), a parameter which combine O:C and H:C, instead of just individually O:C and H:C? Would that reveal clearer trends?

Response: For this, we refer the reader to the response to the comment above this (Comment 18).

20) Line 210ff: if the conditions were so different (namely VOC to oxidant ratio and residence time), are the HOM yields representative for this study?

Response: This issue is addressed in more detail in the response of Comment 10 from Reviewer 1.

21) Generally, HO2 concentrations are very high in PAM when operated at low NOx (very little recycling back to OH). How would that affect HOM production. Don't higher HO2 concentrations enhance the quenching of RO and RO2 radicals and thus suppress auto-oxidation type processes?

Response: The reviewer is correct that $HO_2$ tends to suppress $RO_2$ autoxidation. The RO H-shifts on the other hand are fast enough (sub-millisecond time scales) that bimolecular reactions cannot compete. As such, the main issue would be that the high $HO_2$ would suppress the conversion of $RO_2$ to RO, forming instead ROOH. However, also in the (low-$NO_x$) atmosphere $HO_2$ is a major reaction partner for $RO_2$ radicals, and thus the high $HO_2$ is not making the experiments unrealistic. In addition, the reaction $RO_2 + HO_2 \rightarrow RO + OH + O_2$ is also competitive, and thereby the RO formation is not inhibited even at super high $HO_2$ concentrations.

22) Fig 4: I wonder how useful this figure/comparison is knowing about the limitations of the HOM data. It may be doing more harm than good. Playing devil's advocate, I can look at this figure and state that decanal can show a aerosol yield of 0.08 with molar HOM yield of ~3e-2 and decanone can have the same aerosol yield with "no HOM at all". Thus, HOM cannot be that important for aerosol yields Looking only at the precursors that had a measurable HOM yield, I could claim that and increasing HOM yields show decreasing aerosol yields (decanal& decanol vs decaline and butylcyclohexane).

Response: This issue is addressed in more detail in the response of Comment 9 from Reviewer 1. However, the reviewers' comments about the usefulness/harm of Figure 4 are true. As with the other comments regarding the HOM data, we have now changed the manuscript as described in the response of Major Comment 2 and Comment 10 to Reviewer 1.
Figure 4 is now moved to the Appendix to deemphasize the HOM vs SOA yield correlation as described in the abovementioned responses.

23) Line 237:"and thus high injected VOC concentrations will lead to the majority of OH radicals reacting with the VOC. " I'm not sure if this is strictly true for PAM. A lot of the OH radicals will also react with O3 (which is at 10s of ppm I assume).

Response: The reviewer is correct. Our intention was however to motivate the decreasing trend of $H_n:H_{n+2}$ at higher injection rate with the availability of VOC molecules that OH radicals can react with. This is now rephrased in the revised manuscript as:

Line 283-285: *"This is to be expected, as the OH production stays largely constant, and thus high injected VOC concentrations will lead to OH radicals reacting almost solely with VOC molecules instead of oxidation products.*

24) Line 230ff: Only C10 HOM compounds were investigated for the evaluation of the first/second generation products. Does this interpretation hold when other HOM species are included (e.g. with C9 or C8)?

Response: Although we did observe C8 and C9 compounds, we did not do a full peak identification for other carbon numbers than 10 as our main focus was not on the gas phase data.

25) Line 230ff: Does C-C bond cleavage become more important on successive oxidation? If that is the case, would that not mean that some of the second generation product are no longer C10 and thus "hide" as C<10 HOM? Could this be more pronounced for acyclic compounds?

Response: Yes, acyclic compounds are more prone to fragment during oxidation, and at the $2^{nd}$ OH oxidation step, also cyclic VOCs can be more likely to fragment. Some parts of the reviewers' questions are addressed in Comment 5 from reviewer 1, but the comparison we made here was certainly not quantitative, but more indicative. While the "hiding" effect mentioned by the reviewer is very likely to take place, the fact that we observe it also for purely C10 HOMs suggests that the multi-step oxidation is indeed becoming more important.

26) Line 230ff: This analysis (first/second generation) is only based on the gas phase data. If the multi gen products of decanal etc. are just a bit more low-volatility, they could be condensing into the particle phase, hiding from the gas phase measurements. Then it would look like there is less of them in the gas phase, right?

Response: We refer to our previous response. Also here, the "hiding" is likely taking place, but we nevertheless notice the change also in the gas phase data.

27) Line 273ff: I do not agree with finding "a clear link between the two yields" (see specific comment 22). If anything, the combination of the Wang et al. and this new study shows how much the experimental setup and chosen reaction conditions can impact the formed and detected HOM amounts and types. Thus, great care must be taken in the experiment design and when comparing HOM data from different studies.

Response: See the response of Comment 22 for details. However, we removed the word "clear" from the manuscript, as we agree that the link between HOM and SOA yields are far from clear.

28) Fig A1: This figure needs more explanations in the caption. None of the acronyms/abbreviations are explained. The main text does contain most of the information, but this figure will become much easier to understand if the information is also provided directly with it.

Response: We agree with this and have now added descriptions of all acronyms.

*Figure A1. Experimental setup used in this work. Total flow to the Potential Aerosol Mass (PAM) chamber consisted of (1) a nitrogen (N$_2$) flow where single VOC (volatile organic compound) was injected, (2) humidification flow (through a water bath) and (3) a clean air flow. PAM outflow was diluted with a dilution flow. Particle phase products were measured with a Long Time of Flight Aerosol Mass Spectrometer (AMS) and a Scanning Mobility Particle Sizer (SMPS). Gas phase products were monitored with a nitrate chemical ionisation MS (NO3-CIMS) and a Proton Transfer Reaction ToF MS (PTR-TOF) and an ozone (O3) analyzer.*

**Language**

+ line 11: "emitted in the atmosphere" – should be "emitted into"

Response: corrected as suggested
Line 10*: "...that are emitted into the atmosphere..."*

+ line 112 "got enough of sample flow" – I would omit the "of"

Response: corrected as suggested
Line 146*: "... sure that all instruments got enough sample flow ..."*

+ line 212 "chosen from them" who is "them"? I guess the auhors mean the Wang paper? Please rephrase to make clearer what the "them" is referring to.

Response: corrected, "them" is replaced with "Wang et al. (2021)"
Line 261-262*: "... was chosen from Wang et al. (2021), the ..."*

+ line 235f: The way the increase/decrease trends are assigned in this sentence was a bit hard to wrap my head around. To paraphrase: "decrease of A to B ratio with increasing VOC conc means that lower VOC conc increases second gen oxidation" Consider rephrasing this. "the ratio of second to first generation product ions increased with decreasing VOC conc, i.e., lower precursor conc "

Response: This is now rephrased as:

Line 281-283*: "We indeed observed that the ratio of second to first generation products (i.e., C$_{10}$H$_{14}$O$_y$ to C$_{10}$H$_{16}$O$_y$ (y=4-9)) increased with decreasing injected VOC concentration (Fig. 5), indicating that lower VOC concentration lead to increased second-generation OH oxidation in our system."*

+ line 237: "OH generation" maybe better use "OH production" to differentiate from the other meaning of the word generation (reaction generation) which is used in the same paragraph.

Response: corrected as suggested
Line 283: "... the OH production stays largely constant ..."

---

## Referee Report (RR1)

Supporting Figure for Review Report on Graeffe et al. 2025

[Figure]

Figure 1: Left: part of Figure 1 with added lines to highlight shape of shaded error area. Right: depiction of a symmetrical error in log space for a similar value.

---

## Author Response (AR2)

We thank the reviewer for reviewing the revised manuscript and giving us some more feedback. Below we address the comments point-by-point. The original comments are in black, and our responses are given in red and modifications in the revised manuscript are given in blue.

**Reviewer: 1**

The authors gone to some length to address the comments raised by both reviewers.

My main concern was about the VOC estimation and the related uncertainties. The authors provide good supporting evidence in their reply and added a Appendix Section about the specific and overall uncertainties for all relevant aspects of the SOA yield calculations. While the absence of direct VOC measurements is still not ideal, this is the best that can be done and sufficient for a robust study.

Response: We appreciate the comment from the reviewer.

My second major concern (the interpretation of the HOM results) has been resolved as well by reducing the prominence of the HOM related parts and making it clear that these are only "bonus findings" that support the main topic of the manuscript. That is perfectly fine.

Response: We appreciate the comment from the reviewer.

I found a few mostly technical details in the new sections of the manuscript that the authors should clarify before publication.

1) Why have the formed SOA mass concentrations changed? In Table 1: n-decane old: 1.8-7.3 ug/m3 , new: 4.8-9.3. These are measured values that should not be affected by any of the additional uncertainty calculations and the like. The uuthors state in one of the replies that particle transmission was already corrected in the original manuscript. SO that cannot be the reason for the higher values. What made all these values increase in the revision?

Response: Indeed, the particle transmission was already corrected for in the original manuscript. However, the reason for changed SOA mass concentrations is the new correction from the model for estimating the fate of low-volatility organic compounds (LVOCs) by Palm et al. (2016). As described in Palm et al. (2016), "*The correction, hereafter referred to as the "LVOC fate correction", is applied by dividing the amount of SOA mass formed by $F_{aer}$*" (where $F_{aer}$ is the modelled fraction of LVOCs that are condensed on the aerosol). Therefore, all SOA mass concentrations increased when this correction was applied.
To clarify this, we have now mentioned the LVOC fate correction in the revised manuscript:

Line 137-140: *"*The mass concentration of the SOA was calculated by combining the total particle volume from the SMPS and the SOA density calculated from the elemental ratios. The density was calculated for each step according to the equation in Kuwata et al. (2012), yielding in densities from 1100 to 1400 kg m$^{-3}$. Mass concentration was also corrected by the "LVOC fate correction", according to Palm et al. (2016), as described in the section above."

As we also applied more averaging (for clearer plots and better statistics) in the revised manuscript, i.e., data points with same amount injected VOC are combined to one data point, there will be some changes in the data points plotted in the graphs and written in the table. For example, for n-decane (as the reviewer pointed out), the original the range changed from 1.8-

7.3 ug/m3 to new 4.8-9.3 ug/m3. For the lower end, we have three points (1.83…; 2.61…; 3.17… ug/m3), therefore the lower end of 1.8 ug/m3 in the original manuscript. The modelled $F_{aer}$ for the averaged points is 0.53…, so taking the average of the original SOA mass concentrations and dividing it by $F_{aer}$, we get the new lower end of 4.8 ug/m3. Same thing with the higher end, it has two points (7.27… + 7.06… ug/m3) and $F_{aer}$ is 0.77…, from that we get the new higher end of 9.3 ug/m3.

2) Fig 2: I really like the use of shading in this Figure. But the authors should double check the plotted values. Symmetric errors in linear space (e.g. +/- 30%) will not look symmetrical in log spacing. But some of the shading looks suspiciously symmetrical around the data points. IN the attached document, I picked an example (brown point at ~0.06). I added two black bars of equal length to highlight the similarity of the width of the positive and negative error band. I added an example for 0.06 +/- 40% (red point on the right)

Response: The reviewer is correct; the errors are not symmetrical in Fig 2. We separately calculated the positive and negative errors, by choosing the "worst case scenario" for the two cases. I.e., to find the lower error-band, we minimized the SOA-yield function (that consists of several variables with different uncertainties) and to find the upper error-band, we maximized the SOA-yield function. Thereby, the errors are not symmetrical as the described method for calculating the errors gives different upper and lower values.
This is now also described in the Appendix when describing the uncertainties:

Line 413-417: *"This includes the uncertainty of* both *particle phase instruments, as well as the uncertainty from the LVOC fate model (described in Section 2.1). When calculating the SOA yield errors in Fig. 2, we calculated separately the upper and lower error values. This was done by altering all the variables within their uncertainty ranges in the SOA yield function to find the minimum and maximum values. This method will not result in symmetrical upper and lower errors, as seen in Fig. 2."*

3) Original Specific Comment #1 Line 48 "bimolecular reaction": From the authors reply, I understand what is meant by the phrase "when more bimolecular reactions took place". But looking only at the phrase in the manuscript, it is still not clear which bimolecular reactions are meant in this context. There are many other bimolecular reactions that do not convert RO2 to RO, e.g., the quenching reaction RO2+HO2. The authors need to make this more specific in the manuscript as the general reader will not look at the review replies.

Response: To clarify the paragraph, we have now removed "bimolecular reaction" and rephrased it as:

Line 46-51: *"Wang et al. (2021) did not only measure HOM yields, but showed that the oxygen content in oxidation products generally increased when more* peroxy radicals (RO₂) were converted to alkoxy radicals (RO), *even though not always reaching six or more O-atoms. Much of the O-atom incorporation was attributed to RO₂* reactions *with other RO₂ radicals or NO, forming RO able to isomerize and thus allow reactions with molecular O₂.*

*This is in contrast to many monoterpenes where the RO$_2$ radicals themselves can undergo isomerization reactions (autoxidation), owing to suitable structures in the monoterpene-derived radicals which are less common in alkanes."*

4) Line 240 in Marked Manuscript: "…but as the particle mass increases, more volatile(, less oxidized and longer photochemical aging". Is LONGER photochemical aging really correct here? This is opposite to what is shown in the Figures (E.g. Fig 1 in manuscript). The higher SOA masses were achieved by increasing the VOC concentrations and leaving the oxidant production the same which means that the OH exposure (equivalent photochemical aging time) is lower for the high SOA masses. So it should be "more volatile, less oxidized and SHORTER photochemical aging" here.

Response: The reviewer is correct: "longer" should be replaced with "shorter". This is a mistake from our side. This is also easy to see in Fig. 3b where the colorbar (white to black) shows the photochemical age; we have shorter photochemical age at higher SOA mass concentrations.
This is now corrected for in the revised manuscript as:

Line 241-242: *"…but as the particle mass increases, more volatile (, less oxidized and shorter photochemical aged) compounds can condense…"*
* * *
NOTE: Markes types are modified in Figures 2, 3, 4 and A9 to ensure readers with color vision deficiencies to interpret them correctly. No data is changed, only the visual look is changed.

Kuwata, M., Zorn, S. R., and Martin, S. T.: Using Elemental Ratios to Predict the Density of Organic Material Composed of Carbon, Hydrogen, and Oxygen, Environmental Science & Technology, 46, 787-794, 10.1021/es202525q, 2012.
Palm, B. B., Campuzano-Jost, P., Ortega, A. M., Day, D. A., Kaser, L., Jud, W., Karl, T., Hansel, A., Hunter, J. F., Cross, E. S., Kroll, J. H., Peng, Z., Brune, W. H., and Jimenez, J. L.: In situ secondary organic aerosol formation from ambient pine forest air using an oxidation flow reactor, Atmospheric Chemistry and Physics, 16, 2943-2970, 10.5194/acp-16-2943-2016, 2016.
Wang, Z. D., Ehn, M., Rissanen, M. P., Garmash, O., Quéléver, L., Xing, L. L., Monge-Palacios, M., Rantala, P., Donahue, N. M., Berndt, T., and Sarathy, S. M.: Efficient alkane oxidation under combustion engine and atmospheric conditions, Communications Chemistry, 4, 10.1038/s42004-020-00445-3, 2021.